Subject Areas:
chaos theory/applied mathematics

Keywords:
iterative sequences, random walks, Brownian motion, one-dimensional maps, conditionally invariant densities

Author for correspondence:
Radek Erban
e-mail: erban@maths.ox.ac.uk

# Limiting stochastic processes of shift-periodic dynamical systems

## Julia Stadlmann[1] and Radek Erban[2]

[1]Merton College, Merton Street, Oxford OX1 4JD, UK
[2]Mathematical Institute, University of Oxford, Radcliffe Observatory Quarter, Woodstock Road, Oxford OX2 6GG, UK

RE, 0000-0001-8470-3763

A shift-periodic map is a one-dimensional map from the real line to itself which is periodic up to a linear translation and allowed to have singularities. It is shown that iterative sequences $x_{n+1} = F(x_n)$ generated by such maps display rich dynamical behaviour. The integer parts $\lfloor x_n \rfloor$ give a discrete-time random walk for a suitable initial distribution of $x_0$ and converge in certain limits to Brownian motion or more general Lévy processes. Furthermore, for certain shift-periodic maps with small holes on [0,1], convergence of trajectories to a continuous-time random walk is shown in a limit.

## 1. Introduction

Dynamical systems and their stochastic properties have been studied for more than 100 years, starting with the pioneering works of Poincaré [1], who connected probabilistic concepts with dynamics, conjecturing the Poincaré recurrence theorem. Major advances in the field were made in the 1930s by Birkhoff [2] and von Neumann [3], via the proof of the so-called ergodic theorems, concerning time averages of functions along trajectories. Birkhoff also first used topological methods for the study of dynamical systems. In these early years, differential equations were often the main focus of the study of dynamical systems. However, since the 1970s attention also turned to simple dynamical systems, generated iteratively from a map $F : \Omega \to \Omega$ via an equation

$$x_{n+1} = F(x_n), \qquad (1.1)$$

where $\Omega$ has been taken to be a low-dimensional set [4], such as interval [0,1]. It has been observed that even very simple maps and systems can give rise to complicated, seemingly random behaviour of trajectories, a phenomenon Yorke and Li named 'chaos' in their seminal paper [5]. A well-studied example of this phenomenon is given by the logistic map $F(x; r) = rx(1 - x)$, $x \in [0,1]$, where $r \in (0,4)$ is a parameter. Depending on the value of $r$, it displays a wide array of behaviour of trajectories, highly

sensitive to the initial value [6]. Another interesting function is the climbing sine map, defined by $F(x; a) = x + a \sin(2\pi x)$, $x \in \mathbb{R}$, where $a > 0$ is a parameter. Due to being defined on an unbounded set, it can display diffusive behaviour for large enough values of $a$. Varying its parameter $a$, the dynamics of the climbing sine map can range from localized orbits to ballistic dynamics and chaotic diffusion [7,8].

The climbing sine map is an example of a one-dimensional map displaying a translational symmetry $F(x + 1) = F(x) + 1$ for $x \in \mathbb{R}$. It was discovered in the 1980s that diffusive motion can be generated from purely deterministic systems given by equation (1.1) and certain maps $F$ with the aforementioned symmetry [9]. The climbing sine map $F(x; a)$ itself has been investigated particularly closely; a detailed account of its bifurcation patterns depending on parameter $a$ is given in [10]. Diffusion constants and drift velocity have been studied not just for $F(x; a)$, but also for similar, linearized versions of this model [11,12].

Maps of this type can provide deterministic models of anomalous diffusion (nonlinear growth of mean square displacement) and examples of such systems are linked to another interesting phenomenon, intermittent chaos, whereby long, seemingly periodic behaviour is occasionally interrupted by chaotic bursts [13]. Geisel and his coauthors [14,15] considered maps with the properties $F(x + 1) = F(x) + 1$, $F(-x) = -F(x)$ and $F(x) = (1 + \varepsilon)x + ax^z - 1$ close to 0, where $a = 2^z(1 - \varepsilon/2)$ and $z > 1$ and $\varepsilon > 0$ are parameters. When $z < 2$ normal diffusion occurs, which transitions to anomalous diffusion for $z > 2$. Zumofen & Klafter [16–18] also studied this map, investigating not just the mean square displacement, but also the propagator $P(r, t)$, the probability density to be at $r$ at time $t$, both for non-stationary and stationary initial conditions. Power spectra are analysed in [19]. They establish that the dynamical behaviour can be described well in terms of Lévy walks, which are discussed within a framework of continuous-time random walks, explained in detail in [17].

More recently, diffusion coefficients of some parameter-dependent families of piecewise linear maps with translational symmetry $F(x + 1) = F(x) + 1$ were investigated by Klages & Dorfman [20,21]. They showed that the diffusion coefficient displays a fractal structure as function of a parameter linked to the slopes of the map. Using stronger techniques, such as Taylor–Green–Kubo formulae and generalized Takagi functions, the result was extended to different classes of piecewise linear maps [22], and intermittent maps similar to the Pomeau–Manneville map described earlier [23]. Even the diffusion coefficient of the climbing sine map $F(x; a)$, treated as a function of $a$, has a fractal structure [7,8].

This paper is dedicated to studying stochastic processes generated by repeated application of maps satisfying $F(x + 1) = F(x) + 1$, with a particular focus on the random walk-like behaviour of the resulting trajectories. There seems to be no consensus on a name for such functions, so we will subsequently call them shift-periodic, provided they additionally satisfy a minor technical restriction introduced in Definition 2.1. Although the sequence $(x_n)$ generated from equation (1.1) is fully deterministic for given $x_0$, it can be viewed as a discrete-time stochastic process when its initial value $x_0$ is chosen according to a probability distribution on domain $\Omega$ of the underlying map $F$. A common choice is an invariant distribution with respect to $F$, which allows us to study equilibrium behaviour of trajectories. However, the shift-periodicity of our maps leads to another natural choice of initial distribution, namely one invariant with respect to the fractional parts of $F$ on [0,1]. This is the initial distribution we will work with in most of the paper. We will also consider the behaviour of continuous-time stochastic processes appearing as a limit under a suitable scaling in time and space, explained in §3. Motivation for this is the well-known result that Brownian motion is obtained as a limit after an appropriate scaling for a certain class of maps, as demonstrated by Beck & Roepstorff in [24,25], and further investigated by various other authors, for example by Mackey and Tyran–Kamińska [26,27].

The paper is structured as follows. In §2, we introduce shift-periodic maps using a couple of examples important for later discussions. In §3, we consider a certain class of shift-periodic maps which admit an infinite Markov partition and show that their trajectories behave like discrete-time random walks for initial distributions invariant with respect to the fractional parts of the original map. We also point out conditions on these shift-periodic maps which ensure that certain continuous-time stochastic processes arise in a scaling limit. Finally, in §4, we study shift-periodic maps with small holes and show that in an appropriate scaling limit we obtain the behaviour of a continuous-time random walk.

**Notation.** We denote $\mathbb{R} \cup \{\infty\} \cup \{-\infty\}$ by $\mathbb{R}_\infty$ and $\mathbb{Z} \cup \{\infty\} \cup \{-\infty\}$ by $\mathbb{Z}_\infty$. For any $x \in \mathbb{R}$ let $\{x\}$ denote the fractional part of $x$ and $\lfloor x \rfloor$ denote the integer part of $x$. For any Lebesgue measurable set $A$, we denote its Lebesgue measure by $\lambda(A)$. As it is common in the literature, the symbol $\sim$ will be used in two different contexts. First, for functions $f(x)$ and $g(x)$, we write $f \sim g$ if $f(x)/g(x) \to 1$ as $x \to \infty$. Second, we also use symbol $\sim$ to specify the distribution of a random variable, for example, $X \sim N(0,1)$ means that random variable $X$ is normally distributed with zero mean and unit variance. In §4, we also make use of the sup-norm on the space of bounded functions from [0,1] to $\mathbb{R}$, defined by $\|f\|_\infty = \sup_{x \in [0,1]} |f(x)|$.

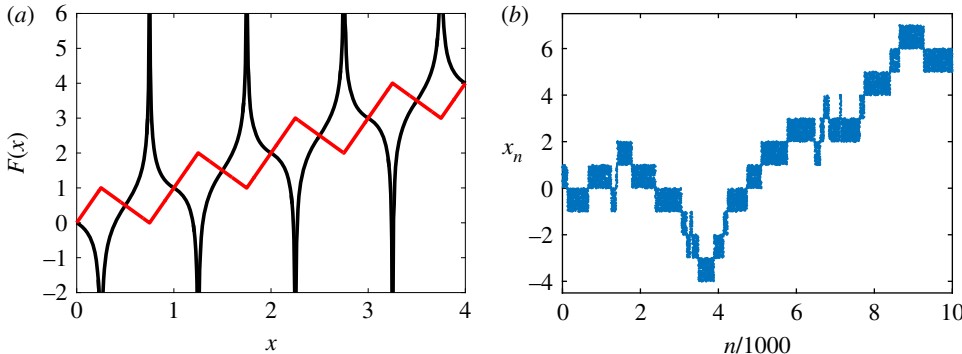

**Figure 1.** (a) Two examples of shift-periodic maps. Continuous piecewise linear map $F(x; \varepsilon, \delta)$ given in example 2.3 for $\delta = \varepsilon = 10^{-2}$ (red solid line) and shift-periodic map with singularities $F(x; \kappa)$ given in example 2.4 for $\kappa = 1$ (black solid line). (b) Illustrative dynamics of shift-periodic map $F(x; \varepsilon, \delta)$ from example 2.3 for $\delta = \varepsilon = 10^{-2}$. The first $10^4$ iterations $x_{n+1} = F(x_n; 10^{-2}, 10^{-2})$ are plotted for initial condition $x_0 = 0.9$.

## 2. Shift-periodic maps

This paper studies the behaviour of iterative sequences given by (1.1) for functions $F$ defined on the real line, which are periodic up to integer shifts. The key property of maps $F$ is a shift-periodic formula given in the next definition as condition (i), together with a minor technical restriction (ii) on discontinuities of $F$. Note that, unlike in other works on this topic in the literature, we allow $F$ to have singularities.

**Definition 2.1.** A shift-periodic map is a map $F : \mathbb{R} \to \mathbb{R}_\infty$ with the following properties:
(i) $F(x) = F(\{x\}) + \lfloor x \rfloor$ for all $x \in \mathbb{R}$;
(ii) There exist $0 = t_0 < t_1 < \cdots < t_k = 1$ so that for $i = 1, 2, \ldots, k$ map $F$ is continuous and monotonic on $(t_{i-1}, t_i)$.

**Example 2.2.** In §1, we discussed the climbing sine map $F(x; a) := x + a \sin(2\pi x)$ as an example of a map with translational symmetry possessing interesting dynamical properties. It satisfies the conditions of definition 2.1. More generally, a wide variety of shift-periodic maps can be constructed by choosing two polynomials $P(x)$ and $Q(x)$, the latter non-zero, and setting $F(x) = x + P(\sin(2\pi x))/Q(\cos(2\pi x))$. For example, take $P(x) = Q(x) = x$ to get $F(x) = x + \tan(2\pi x)$. In figure 2b, a sample trajectory of this map is plotted in red.

The nonlinearity of the climbing sine map and other climbing trigonometric functions however complicates the discussion of invariant densities and the behaviour of iterates in general. So below, in example 2.3, we discuss a piecewise linear map $F(x; \varepsilon, \delta)$ of a similar structure.

**Example 2.3.** We consider a piecewise linear map $F : \mathbb{R} \to \mathbb{R}$ with parameters $\delta > 0$ and $\varepsilon > 0$ which satisfies $F(x; \varepsilon, \delta) = F(\{x\}; \varepsilon, \delta) + \lfloor x \rfloor$ for $x \in \mathbb{R}$ and is defined on [0,1] by

$$
F(x; \varepsilon, \delta) := \begin{cases}
(4 + \varepsilon)x, & \text{if } x \in \left[0, \frac{1}{4}\right); \\
(-2 - \varepsilon)x + \frac{(3+\varepsilon)}{2}, & \text{if } x \in \left[\frac{1}{4}, \frac{1}{2}\right); \\
(-2 - \delta)x + \frac{(3+\delta)}{2}, & \text{if } x \in \left[\frac{1}{2}, \frac{3}{4}\right); \\
(4 + \delta)x - (3 + \delta), & \text{if } x \in \left[\frac{3}{4}, 1\right].
\end{cases}
$$

The map $F(x; \varepsilon, \delta)$ has one local maximum and one local minimum in interval [0,1], maps interval [0,1] to a larger interval $[-\delta/4, 1 + \varepsilon/4]$ for parameters $\delta > 0$ and $\varepsilon > 0$, and is plotted in figure 1a (as a red solid line). Behaviour of its iterates (1.1) closely resembles that of a random walk. Choosing relatively small values $\delta = \varepsilon = 10^{-2}$, the first $10^4$ iterations of map from example 2.3 are shown in figure 1b. Identifying intervals $[i, i + 1)$ with integer valued lattice points $\{i\}$ for $i \in \mathbb{Z}$, we observe that sequence $x_n$ can be viewed as a random walk between these lattice points. More precisely, we can map sequence $x_n$ to an integer-valued sequence $\lfloor x_n \rfloor$, which gives lattice positions of a random walker that is jumping from site $\{i\}$ to neighbouring sites $\{i - 1\}$ and $\{i + 1\}$ with certain probabilities. Such behaviour is common among trajectories of shift-periodic

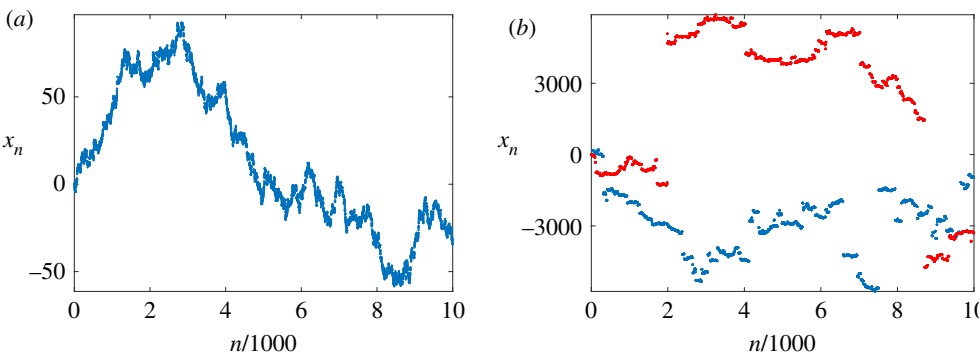

**Figure 2.** Illustrative dynamics of shift-periodic map $F(x; \kappa)$ from example 2.4. The first $10^4$ iterations $x_{n+1} = F(x_n; \kappa)$ are plotted (*a*) for $\kappa = 10$ and initial condition $x_0 = 0.4$; (*b*) for $\kappa = 1$ and initial condition $x_0 = 0.2$ (blue dots). Panel (*b*) additionally includes a plot of a sample trajectory of map $F(x) = x + \tan(2\pi x)$, with $x_0 = 0.2$ (red dots).

maps and under certain conditions on $F$ the jumps between sides are independent, as is traditionally required of random walks. This will be discussed in §3. The map in example 2.3 does not satisfy this independence condition, but attains the structure of a continuous-time random walk with independent waiting times in a suitable limit. Section 4 is dedicated to this result.

Finally, a more general example of a shift-periodic map, in the spirit of the climbing tangent map, is illustrated in figure 1*a* as a black solid line and is formally defined in example 2.4.

**Example 2.4.** We consider $F : \mathbb{R} \to \mathbb{R}_\infty$ with parameter $\kappa > 0$, defined on [0,1] by

$$F(x; \kappa) = \frac{4^{-1/\kappa}}{2(1 - 3^{-1/\kappa})} \left( \left| x - \frac{3}{4} \right|^{-1/\kappa} - \left| x - \frac{1}{4} \right|^{-1/\kappa} \right) + \frac{1}{2},$$

and with $F(x; \kappa) = F(\{x\}; \kappa) + \lfloor x \rfloor$ for $x \in \mathbb{R}$. The prefactor is chosen so that $F(0; \kappa) = 0$ and $F(1; \kappa) = 1$.

In figure 2, we plot illustrative trajectories for two different values of $\kappa$. For large $\kappa$ (figure 2*a*), the behaviour of iterations $x_{n+1} = F(x_n; \kappa)$ resembles Brownian motion, while for small $\kappa$ (figure 2*b*, blue dots) it resembles a Lévy flight. We write 'resembles' since we compare discrete dynamics with continuous time stochastic processes. In §3.6, we make these statements rigorous. To do this, we identify the index $n$ in $x_n$ with time and introduce suitable scaling of time to get convergence to a continuous-time process. Before that, we study the random walk behaviour of a certain class of shift-periodic maps when time is left unscaled.

# 3. Discrete-time random walks

While the iterative formula (1.1) uniquely determines the next iterate $x_{n+1}$ from the knowledge of $x_n$, figures 1*b* and 2 suggest that the next value $\lfloor x_{n+1} \rfloor$ is determined from $\lfloor x_n \rfloor$ only with a certain probability. The goal of this section is to formalize this observation for certain shift-periodic maps by studying the connections between the dynamics of (1.1) and random walks, defined below.

**Definition 3.1.** Let $(Y_n)_{n \in \mathbb{N}}$ be a discrete-time stochastic process and let $Z_n = Y_n - Y_{n-1}$ for $n \in \mathbb{N}$, where we assume $Y_0 = 0$. We say $(Y_n)_{n \in \mathbb{N}}$ is a discrete-time random walk if $Z_n$, $n \in \mathbb{N}$, are independent and identically distributed.

Now, as briefly noted in §2, the increments $\lfloor x_n \rfloor - \lfloor x_{n-1} \rfloor$ of sequence $\lfloor x_n \rfloor$, $n \in \mathbb{N}$, are not generally independent. For instance, if the parameters $\delta$ and $\varepsilon$ of map $F(x; \varepsilon, \delta)$ in example 2.3 are chosen sufficiently small, consecutive jumps between unit intervals $[i, i+1)$, $i \in \mathbb{Z}$, are not possible. A key issue here is that the local extrema of $F(x; \varepsilon, \delta)$ do not take integer values. This motivates the next definition, introducing a restriction on shift-periodic maps for which an appropriate distribution of the initial values guarantees that the behaviour of a sequence generated by such a shift-periodic map will be that of a random walk.

**Definition 3.2.** Let $F$ be a shift-periodic map with $0 = t_0 < t_1 < \cdots < t_k = 1$ such that $F$ is continuous and monotonic on $(t_{i-1}, t_i)$. We then say that $F$ has integer spikes if $F$ additionally satisfies conditions (iii) and (iv) below:

(iii) $|F(x) - F(y)| > |x - y|$ holds for all distinct $x, y \in (t_{i-1}, t_i)$ where $i = 1, 2, \ldots, k$.

and

(iv) $\lim_{x \to t_i^+} F(x) \in \mathbb{Z}_\infty$ and $\lim_{x \to t_i^-} F(x) \in \mathbb{Z}_\infty$ for $i = 0, 1, 2, \ldots, k$.

Condition (iii) is a technical restriction, which is sometimes called expanding in the literature [28], although this terminology is usually reserved for stronger conditions [29]. It will be important in lemma 3.7 later. Condition (iv) ensures that $\{F\}$ admits a Markov partition on [0,1], a partition of the unit interval into a collection of subintervals such that the fractional parts of $F$ map any such interval onto a union of other intervals in the partition. Precisely, there exist disjoint intervals $(a_i, b_i)$, $i \in I$, where $I$ is a countable indexing set, such that $[0, 1] \setminus \cup_{i \in I}(a_i, b_i)$ is countable and for any $i \in I$ there exists $J \subseteq I$ with $\{F((a_i, b_i))\} = \cup_{i \in J}(a_j, b_j)$. The intervals can be chosen so that $F$ is continuous and monotonic on each of them. This ensures that each trajectory of the associated dynamical system can be described by a sequence of symbols corresponding to the different intervals, and often significantly simplifies the analysis of dynamical behaviour, particularly in the case of finite Markov partitions. For instance, the invariant density of piecewise linear maps admitting a finite Markov partition is easy to compute since its Frobenius–Perron operator can be described in matrix form [29]. In this section, we allow infinite partitions which will play an important role in the proof of lemma 3.7, where we will make use of the aforementioned representation of trajectories in terms of sequences of intervals.

**Theorem 3.3.** *Let $F : \mathbb{R} \to \mathbb{R}_\infty$ be a shift-periodic map with integer spikes and let $U$ be uniformly distributed on [0,1]. Then there exists a homeomorphism $h : [0,1] \to [0,1]$ so that for $Y_n = \lfloor F^n(h(U)) \rfloor$ and $Y_0 = 0$, the stochastic process $(Y_n)_{n \in \mathbb{N}}$ is an integer-valued discrete-time random walk and we have $\mathbb{P}(Y_n - Y_{n-1} = m) = p_m$, where $p_m$ is a constant satisfying*

$$p_m = \lambda(\{x \in [0, 1] : \lfloor F(h(x)) \rfloor = m\}) \qquad \text{for any } m \in \mathbb{N}.$$

We take two different approaches to proving this statement. First, we consider shift-periodic maps $F$ for which the fractional parts on [0,1] have an absolutely continuous invariant measure $\mu$. Under certain conditions on the map $h$ defined by $h^{-1}(x) = \mu([0,x])$, theorem 3.3 will be satisfied, see proposition 3.5. Then we describe a second way to construct a suitable homeomorphism $h : [0,1] \to [0,1]$, valid for any choice of shift-periodic map $F$ with integer spikes. Our construction will ensure $\lambda(\{x \in [0, 1] : \lfloor F(h(x)) \rfloor = m\}) = \lambda(\{x \in [0, 1] : \lfloor F(x) \rfloor = m\})$ for any $m \in \mathbb{Z}$, so that the transition probability $p_m$ equals the length of the intervals on which $F$ has integer part $m$. Caution must be taken with the interpretation of the latter result, as here the measure of $h(U)$ might not be absolutely continuous with respect to the Lebesgue measure. In that case, typical trajectories might not display the discussed random walk structure.

To investigate random walk behaviour, it will be beneficial to split the map into its fractional and integer parts, considering it as a skew product of maps on [0,1] and $\mathbb{Z}$. We will introduce this concept in the next subsection.

## 3.1. Skew-products

Let $F$ be a shift-periodic map. We first define the 'restricted map', $F_r : [0,1] \to [0,1]$, given by

$$F_r(x) = \begin{cases} \{F(x)\}, & \text{if } F(x) \notin \{-\infty, \infty\}; \\ 0, & \text{if } F(x) \in \{-\infty, \infty\}. \end{cases}$$

We now define a map $\phi_F : [0,1] \times \mathbb{N} \to \mathbb{Z}$ by

$$\phi_F(x, n) = \sum_{k=0}^{n-1} \lfloor F(F_r^k(x)) \rfloor, \tag{3.1}$$

whenever $\lfloor F(F_r^k(x)) \rfloor \notin \{-\infty, \infty\}$ for any $k \in \{0, 1, \ldots, n-1\}$. Else we take $\phi_F(x, n) = 0$. We call $\phi_F$ the cocycle associated with $F$. Note that the conditions placed on shift-periodic map $F$, in particular (i), ensure that whenever $\{F(x), F^2(x), \ldots, F^n(x)\}$ does not intersect with $\{\infty, -\infty\}$ then $F_r^n(x) = \{F^n(x)\}$ and

$$F^n(x) = F_r^n(x) + \phi_F(x, n) \quad \text{for } n \geq 1. \tag{3.2}$$

So for sequence $x_n$ defined by iterative formula (1.1) with starting value $x_0 \in [0,1]$, we have $\lfloor x_n \rfloor = \phi_F(x_0, n)$, provided $F^n(x_0)$ is never infinite. Condition (ii) of definition 2.1 ensures that this

description of the sequence is valid away from a set of Lebesgue measure 0. We can rephrase theorem 3.3 in the following way:

**Theorem 3.3.\*** *Let $F : \mathbb{R} \to \mathbb{R}_\infty$ be a shift-periodic map with integer spikes and let $U$ be uniformly distributed on $[0,1]$. Then there exists a homeomorphism $h : [0,1] \to [0,1]$ so that for $Y_n = \phi_F(h(U), n)$ and $Y_0 = 0$, the stochastic process $(Y_n)_{n \in \mathbb{N}}$ is an integer-valued discrete-time random walk and we have $\mathbb{P}(Y_n - Y_{n-1} = m) = p_m$, where $p_m$ is a constant satisfying*

$$p_m = \lambda(\{x \in [0,1] : \phi_F(h(x), 1) = m\}) \quad \text{for any } m \in \mathbb{N}. \tag{3.3}$$

**Remark.** Cocycle $\phi_F(x, n)$ allows us to associate with $F$ a skew-product $F_s$ induced by cocycle $\phi_F$. Let $F_s : [0,1] \times \mathbb{Z} \to [0,1] \times \mathbb{Z}$ and

$$F_s(x, m) = (F_r(x), m + \lfloor F(x) \rfloor) = (F_r(x), m + \phi_F(x, 1)).$$

Using equation (3.1), $F_s$ satisfies $F_s^n(x, m) = (F_r^n(x), m + \phi_F(x, n))$ for $x \in [0,1]$. Skew-products in general frequently appear as models in physics, for example, see [30] for a recent discussion of diffusion and Lévy-type behaviour of different systems from the point of view of skew-products, including a version of the Pomeau–Manneville map [13] mentioned in §1. Skew-products are also used to investigate recurrence of random walks in [31,32]. Later, in the proof of lemma 3.7, we will relate the trajectories of piecewise monotone maps to those of piecewise linear maps via conjugacy. Similarly, in [33], skew-products with piecewise monotone fibres are related to those with piecewise linear fibres via a semiconjugacy, though in [33] the map is only allowed to have a finite number of monotonic branches. For more general discussions of cocycles and skew-products see [34].

## 3.2. Piecewise linear maps

Theorem 3.3.\* is easy to verify for maps which are linear in between integer function values. (In this case, map $h$ can simply be taken to be the identity map.) What we mean by 'linear between integer values' is made precise in lemma 3.4.

**Lemma 3.4.** *Let $F : \mathbb{R} \to \mathbb{R}_\infty$ be a shift-periodic map with integer spikes and let $U$ be uniformly distributed on $[0,1]$. Suppose there exists a collection of open, pairwise disjoint intervals $\{(a_i, b_i) : i \in I\}$, $I$ countable, so that $[0,1] \setminus \bigcup_{i \in I} (a_i, b_i)$ is countable, $F_r$ is linear on $(a_i, b_i)$ and $F_r((a_i, b_i)) = (0,1)$. Then theorem 3.3 holds for $h(x) = x$.*

*Proof of lemma 3.4.* Since $F$ has integer spikes, $F_r$ linear on $(a_i, b_i)$ with $F_r((a_i, b_i)) = (0,1)$ implies that $\phi_F(x, 1) = \lfloor F(x) \rfloor$ is constant on $(a_i, b_i)$ for each $i \in I$. For $m \in \mathbb{Z}$, we set $I_m = \{i \in I : \phi_F(x, 1) = m$ on $(a_i, b_i)\}$. Then $p_m = \lambda(\bigcup_{i \in I_m} (a_i, b_i)) = \sum_{i \in I_m} (b_i - a_i)$.

Let $Z_1 = \phi_F(U, 1)$ and $Z_j = \phi_F(U, j) - \phi_F(U, j-1) = \phi_F(F_r^{j-1}(U), 1)$ for $j \geq 2$. We now wish to show that for any $m_1, \ldots, m_n \in \mathbb{Z}$ we have $\mathbb{P}(Z_1 = m_1, \ldots, Z_n = m_n) = p_1 \ldots p_n$. For $n = 1$, the statement holds by definition of $p_m$. For $n > 1$ note that $F_r$ is linear on $(a_i, b_i)$ with $F_r((a_i, b_i)) = (0,1)$, so $F_r(U)$ is uniformly distributed on the unit interval conditional on $U \in (a_i, b_i)$. Then $\mathbb{P}(Z_2 = m_2, \ldots, Z_n = m_n \mid U \in (a_i, b_i)) = \mathbb{P}(Z_1 = m_2, \ldots, Z_{n-1} = m_n)$.

The statement then follows by induction. This gives us independence of random variables $Z_n$, $n \in \mathbb{N}$, and further $\mathbb{P}(Z_n = m) = p_m$. The lemma holds. ∎

## 3.3. Absolutely continuous invariant measures

As mentioned in our earlier discussion, it is sometimes possible to deduce the behaviour of trajectories of a shift-periodic map $F$ from a given absolutely continuous invariant measure $\mu$ of $F_r$. Since $\mu$ is absolutely continuous, $x \mapsto \mu([0, x])$ is continuous, and if this map is additionally strictly increasing, it has an inverse denoted by $h$. Crucial now is the integer spike condition: we may split the unit interval into subintervals $(a_i, b_i)$, $i \in I$ on which $F_r$ is monotonic with $F_r((a_i, b_i)) = (0,1)$. If $h^{-1} \circ F_r \circ h$ is additionally linear on $h^{-1}((a_i, b_i))$, our earlier lemma 3.4 becomes applicable.

**Proposition 3.5.** *Let $F : \mathbb{R} \to \mathbb{R}_\infty$ be a shift-periodic map with integer spikes and let $U$ be uniformly distributed on $[0,1]$. Let $(a_i, b_i)$, $i \in I$ be intervals such that $F_r$ is monotonic on $(a_i, b_i)$ with $F_r((a_i, b_i)) = (0,1)$ and $[0,1] \setminus \bigcup_{i \in I} (a_i, b_i)$ is countable. Suppose that $\mu$ is an absolutely continuous invariant measure for $F_r$ and that the map $x \mapsto \mu([0, x])$ is strictly increasing on $[0,1]$. Let $h$ be the inverse of $x \mapsto \mu([0, x])$. Suppose that $h^{-1} \circ F_r \circ h$ is linear on each $h^{-1}((a_i, b_i))$, $i \in I$. Then $Y_n = \phi_F(h(U), n)$, where $n \in \mathbb{N}$, gives a random walk with transition probabilities given by equation (3.3).*

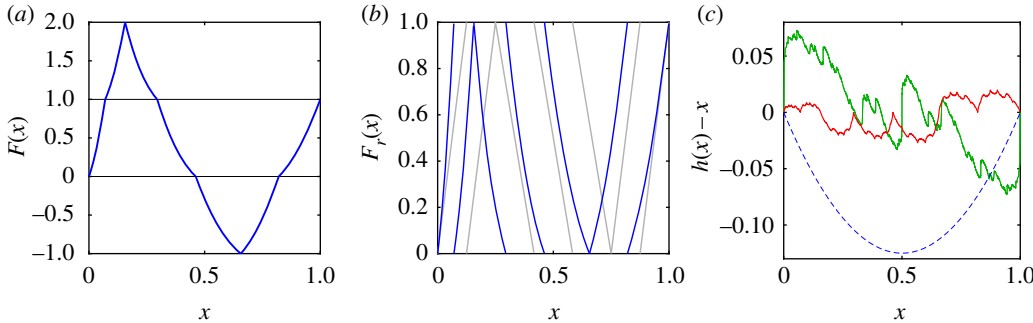

**Figure 3.** (a) Map $F(x)$ defined in equation (3.4) for $h(x) = x(1+x)/2$. (b) Corresponding fractional parts $F_r(x)$ (blue) and map $G_r(x)$ (grey) from equation (3.4). (c) Function $h(x) - x$ for $h$ constructed as in lemma 3.7 for the map in (a), plotted as a red solid line. The blue dashed line shows $x(1+x)/2 - x$ for comparison. The green solid line shows $h(x) - x$ for a version of the Pomeau–Manneville map, given by $F(x) = x + 6x^2 - 1$ on $[0,1/2)$.

*Proof.* Define a shift-periodic map $G : \mathbb{R} \to \mathbb{R}_\infty$ by $G(x) = (h^{-1} \circ F_r \circ h)(x) + \lfloor F(h(x)) \rfloor$. Then $\lfloor G(x) \rfloor = \lfloor F(h(x)) \rfloor$ implies that $\lfloor F(F_r^{n-1}(h(x))) \rfloor = \lfloor F(h(G_r^{n-1}(x))) \rfloor = \lfloor G(G_r^{n-1}(x)) \rfloor$ and so $\phi_F(h(x), n) = \phi_G(x, n)$. Applying lemma 3.4 to $G$, we note that $Y_n = \phi_G(U, n) = \phi_G(h(U), n)$ is a random walk with transition probabilities $p_m = \lambda(\{x \in [0,1] : \phi_G(x, 1) = m\}) = \lambda(\{x \in [0,1] : \phi_F(h(x), 1) = m\})$. ∎

In figure 3a, we present an example of a map $F$ which satisfies the conditions of proposition 3.5. It is constructed by conjugating the piecewise linear map $G(x) = F(x; 4, 4)$, given in example 2.3, with homeomorphism $h(x) = x(1+x)/2$. More precisely, we take

$$F(x) = (h \circ G_r \circ h^{-1})(x) + \lfloor G(h^{-1}(x)) \rfloor. \tag{3.4}$$

Its invariant distribution, corresponding to an absolutely continuous invariant measure, is $h(U)$. Although, starting from a piecewise linear map and a given $h$, we can use equation (3.4) to construct many other examples satisfying proposition 3.5, the statement of theorem 3.3 is more general, as we instead need to construct $h$.

## 3.4. Topological conjugacy

We will now relate the trajectories generated by an arbitrary shift-periodic map $F$ with integer spikes to those of a shift-periodic map $G$ which is linear between integer function values. Below is a formal definition of topological conjugacy, a construction which has already appeared in proposition 3.5 and in equation (3.4).

**Definition 3.6.** Maps $f : [0,1] \to [0,1]$ and $g : [0,1] \to [0,1]$ are topologically conjugate if there exists a homeomorphism $h : [0,1] \to [0,1]$ such that $g = h^{-1} \circ f \circ h$.

Baldwin [28] describes all classes of topologically conjugate maps on $[0,1]$ which are continuous and piecewise monotonic. In [28,33], it is assumed that piecewise monotonic maps have only finitely many monotonic branches. In [35], conjugates between maps with infinitely many monotonic branches are considered, though piecewise differentiability is assumed. A slight adaptation of Baldwin's proof establishes the Lemma below, which is valid for maps with infinitely monotonic branches and does not require differentiability.

**Lemma 3.7.** *Let $f : [0,1] \to [0,1]$ and suppose there exist pairwise disjoint intervals $(a_i, b_i)$ with $i \in I$, where $I$ is a countable indexing set disjoint from $[0,1]$, such that the following conditions are satisfied:*

(i) *For any $i \in I$, $f$ is continuous and monotonic on $(a_i, b_i)$ with $f((a_i, b_i)) = (0, 1)$;*
(ii) *Inequality $|f(x) - f(y)| > |x - y|$ holds for all distinct $x, y \in (a_i, b_i)$, where $i \in I$.*
(iii) *Set $\Lambda = [0, 1] \setminus \cup_{i \in I}(a_i, b_i)$ is countable and $f(\Lambda) = \{0\}$.*

*Define a corresponding linearized map $g : [0,1] \to [0,1]$ in the following way: $g$ is linear on $(a_i, b_i)$ with $\lim_{x \to a_i^+} g(x) = \lim_{x \to a_i^+} f(x)$ and $\lim_{x \to b_i^-} g(x) = \lim_{x \to b_i^-} f(x)$ for any $i \in I$ and further $g(x) = 0$ for any $x \in \Lambda$. Then $f$ and $g$ are topologically conjugate.*

*Proof.* With any $x \in [0,1]$ we associate a sequence $\mathbf{a}_f(x) = (a_0(x), a_1(x), \ldots)$ where $a_n(x) = i$ if $f^n(x) \in (a_i, b_i)$ and $a_n(x) = f^n(x)$ if $f^n(x) \in \Lambda$. Let $\mathbf{b}$ be a finite sequence of length $n$. For sequence $\mathbf{c}$, infinite or finite of

length greater or equal $n$, we say $\mathbf{c}|_n = \mathbf{b}$ if the first $n$ entries of $\mathbf{b}$ and $\mathbf{c}$ coincide. For a finite sequence $\mathbf{b}$ of length $n$ with entries in $I \cup \Lambda$ then write

$$J_{\mathbf{b}}^f = \{x \in [0, 1] : \mathbf{a}_f(x)|_n = \mathbf{b}\}.$$

We now make the following observations: suppose $x \neq y$. Because of condition (ii) it is clear that for some $n \in \mathbb{N}$ the interval $(f^n(x), f^n(y))$ or $(f^n(y), f^n(x))$ intersects with $\Lambda$. We then easily deduce $\mathbf{a}_f(x) \neq \mathbf{a}_f(y)$.

Let again $\mathbf{b}$ denote a finite sequence. From above we then note that $J_{\mathbf{b}}^f$ is empty or a singleton if at least one of the entries of $\mathbf{b}$ is in $\Lambda$. Otherwise $J_{\mathbf{b}}^f$ is an open interval. This can easily be shown by induction, and is a consequence of $f((a_i, b_i)) = (0, 1)$ and of monotonicity of $f$ on $(a_i, b_i)$. Furthermore, for another finite sequence $\mathbf{c}$, if $\mathbf{c}|_n = \mathbf{b}$ then $J_{\mathbf{c}}^f \subseteq J_{\mathbf{b}}^f$. The same observations also apply to $g$, where we define $J_{\mathbf{b}}^g = \{x \in [0, 1] : \mathbf{a}_g(x)|_n = \mathbf{b}\}$. The construction of $g$, in particular equality to $f$ on $\Lambda$, ensures that $J_{\mathbf{b}}^g$ is empty, a singleton or open interval if and only if $J_{\mathbf{b}}^f$ is empty, a singleton or an open interval, respectively.

The sets $J_{\mathbf{b}}^f$, where $\mathbf{b}$ are sequences of length $n$ in $I \cup \Lambda$, partition $[0,1]$. Now we may define a continuous, monotonically increasing map $h_n : [0,1] \to [0,1]$ by

$$h_n(J_{\mathbf{b}}^g) = J_{\mathbf{b}}^f \quad \text{and} \quad h_n \text{ linear on } J_{\mathbf{b}}^f, \tag{3.5}$$

where $\mathbf{b}$ runs over all finite sequences of length $n$ in $I \cup \Lambda$. Note that the least upper bound on the lengths of intervals $J_{\mathbf{b}}^f$, where $\mathbf{b}$ sequence with $n$ entries, goes to 0 as $n$ goes to infinity. $h_n$ thus converges uniformly to a continuous, monotonically increasing map $h$. It is in fact strictly increasing: Note that for $x < y$ there must exist $z_1, z_2 \in [0,1]$ with $x < z_1 < z_2 < y$ so that $\mathbf{a}_g(z_1)$ and $\mathbf{a}_g(z_2)$ have some entries contained in $\Lambda$. This implies that $(h_n(z_1))$ and $(h_n(z_2))$ have constant tails and so, as $h_n(x) < h_n(z_1) < h_n(z_2) < h(z_2)$, taking the limit as $n \to \infty$ we obtain $h(x) \leq h(z_1) < h(z_2) \leq h(y)$.

Now consider $\mathbf{b} = \mathbf{a}_g(x)|_n$ for some $n \in \mathbb{N}$. Since $h$ maps $x$ to $\overline{J_{\mathbf{b}}^f}$ for each such $\mathbf{b}$, $h(x)$ must either have $\mathbf{a}_g(x) = \mathbf{a}_f(h(x))$ or $\mathbf{a}_f(h(x))$ intersects with $\Lambda$. Assume the latter holds. There is a point $y$ with $\mathbf{a}_g(y) = \mathbf{a}_f(h(x))$ and $h_n$ is eventually constant and equal $h$ at $y$. Then $\mathbf{a}_g(y)$ and $\mathbf{a}_f(h(y))$ agree up to arbitrary length and $\mathbf{a}_g(y) = \mathbf{a}_f(h(y))$. But $\mathbf{a}_f(h(x)) = \mathbf{a}_g(y) = \mathbf{a}_f(h(y))$ contradicts $h$ being strictly increasing, unless $x = y$. So $\mathbf{a}_g(x) = \mathbf{a}_f(h(x))$

As $h$ is continuous and strictly increasing, it is also a homeomorphism on $[0,1]$ with $\mathbf{a}_g(x) = \mathbf{a}_f(h(x))$. Then also $\mathbf{a}_g(g(x)) = \mathbf{a}_f(f(h(x)))$ and so $\mathbf{a}_g(g(x)) = \mathbf{a}((h^{-1} \circ f \circ g)(x))$. Therefore, $(h^{-1} \circ f \circ h)(x) = g(x)$ and thus $f$ and $g$ are topologically conjugate. ∎

While the topological conjugacy described above sends $f$ to the linearization $g$ which keeps the endpoints of monotonic branches fixed, the construction itself is valid for any map $g$ which has the same number and orientation of monotonic branches as $f$, or, for an $f$ with infinitely many branches, when $g$ has the same number of limit points of $[0, 1] \backslash \cup_{i \in I} (a_i, b_i)$, corresponding to singularities.

## 3.5. Proof of theorem 3.3

The proof of theorem 3.3 now follows from the previous results: Let $U$ be uniformly distributed on $[0,1]$. Let $F : \mathbb{R} \to \mathbb{R}_\infty$ be a shift-periodic map with integer spikes. The conditions placed on $F$, in particular integer spikes, ensure that $F_r$ satisfies the requirements of lemma 3.7. Let $g$ be the corresponding linearized map, so that $F_r$ is topologically conjugate to $g$. Define a shift-periodic map $G : \mathbb{R} \to \mathbb{R}_\infty$ by setting $G(x) = \lfloor F(x) \rfloor + g(x)$ on $[0,1)$, so that $G_r(x) = g(x)$. Then $G_r = h^{-1} \circ F_r \circ h$ for some homeomorphism $h$.

Next note that $\lfloor F(x) \rfloor = \lfloor G(x) \rfloor$ on $[0,1]$ and that $\lfloor F(x) \rfloor$ is constant on each of the intervals $(a_i, b_i)$ defined for $f = F_r$ in lemma 3.7. Further, by construction, $h((a_i, b_i)) = (a_i, b_i)$. These observations then show

$$\lfloor F(F_r^{n-1}(h(x))) \rfloor = \lfloor G(F_r^{n-1}(h(x))) \rfloor = \lfloor G(h(G_r^{n-1}(x))) \rfloor = \lfloor G(G_r^{n-1}(x)) \rfloor.$$

So $\phi_F(h(x), n) = \phi_G(x, n)$. But $G$ satisfies the conditions of lemma 3.4, so random variables $\phi_G(U, n)$, $n \in \mathbb{N}$, form a random walk with transition probabilities

$$p_m = \lambda\{x \in [0, 1] : \phi_G(x, 1) = m\} = \lambda\{x \in [0, 1] : \phi_F(x, 1) = m\} = \lambda\{x \in [0, 1] : \phi_F(h(x), 1) = m\}.$$

Hence the same must be true for $Y_n = \phi_F(h(U), n)$, $n \in \mathbb{N}$ and theorem 3.3 holds.

**Corollary 3.8.** *Let $U$ be distributed uniformly on $[0,1]$. Let $h$ be the map from lemma 3.7. Then $h(U)$ is an invariant distribution with respect to $F_r$, meaning that for any $x \in \mathbb{R}$*

$$\mathbb{P}(F_r(h(U)) \leq x) = \mathbb{P}(h(U) \leq x).$$

*Proof.* Let $G$ be as in the proof of theorem 3.3. Since $G_r$ is linear on each $(a_i, b_i)$ and $h((a_i, b_i)) = (0, 1)$, we see that $U$ is an invariant distribution with respect to $G_r$. So we deduce $\mathbb{P}(F_r(h(U)) \leq x) = \mathbb{P}(h(G_r(U)) \leq x)$ $\mathbb{P}(G_r(U) \leq h^{-1}(x)) = \mathbb{P}(U \leq h^{-1}(x)) = \mathbb{P}(h(U) \leq x)$. ∎

As noted below the proof of lemma 3.7, the choice of $h$ in the proof of theorem 3.3 is rather arbitrary in the sense that there are infinitely many piecewise linear maps which $F_r$ conjugates to, and for any of these maps, theorem 3.3 is satisfied. The choice of $h$ in lemma 3.7 has the nice property that $p_m = \lambda(\{x \in [0, 1] : \lfloor F(x) \rfloor = m\})$. However, $h(U)$ is not necessarily absolutely continuous, and neither is this necessarily the case for any of the alternative conjugacies.

As an example, consider again $F(x)$ defined in equation (3.4) with $h(x) = x(1 + x)/2$. We know that $h$ is a smooth map which conjugates $F_r$ to the piecewise linear map $G_r$. The map $h(x) - x = x(x - 1)/2$ is plotted in figure 3c as a blue dashed line. However, the construction in lemma 3.7 gives a much more complicated choice of $h$, shown in figure 3c in red. For this choice of $h$, the measure corresponding to distribution $h(U)$ is no longer absolutely continuous with respect to the Lebesgue measure, and the results of theorem 3.3 do not translate into observations of typical trajectories.

The red line in figure 3c illustrates an example in which lemma 3.7 fails to give us a more natural choice of $h$, which exists for this example (blue dashed line). For other shift-periodic maps, however, there is no choice of $h$ for which the measure corresponding to $h(U)$ is absolutely continuous. An example is the Pomeau–Manneville map introduced in §1. Choosing its parameters so that $F(x) = x + 6x^2$ on $[0, 1/2)$, it satisfies the conditions of theorem 3.3, but has derivative equal to 1 at $x = 0$ and $x = 1$. Whenever a trajectory comes close to one of these points, a long sequence of constant integer parts follows until jumps reoccur. The long-term dynamics of a typical trajectory do not display the discussed random walk structure. The map $h$ constructed in lemma 3.7 is shown in figure 3 as a green line.

## 3.6. Alpha-stable processes

The results in the previous subsection showed how iterates $Y_n = \lfloor F^n(X) \rfloor$ of shift-periodic maps with integer spikes can be viewed as sums of independent and identically distributed random variables $Y_n - Y_{n-1}$, for suitable choices of initial distribution $X$. A lot is known about the behaviour of continuous stochastic processes arising as the limit of such partial sum processes under suitable scaling, for a summary see for example [36]. We will briefly give an overview of such results for independent random variables and the implications for our shift-periodic maps.

First, we make our notion of a scaling, passing from a discrete to a continuous process, precise. Take $Y_n = \lfloor F^n(h(U)) \rfloor$, choose translation-scaling and space-scaling constants $a_n$ and $b_n$, and set

$$V^{(n)}(t) = \frac{1}{b_n}(Y_{\lfloor nt \rfloor} - a_n t), \quad \text{where } n \in \mathbb{N}. \tag{3.6}$$

Since for integer spike maps as in theorem 3.3 the generated random variables $Y_{\lfloor nt \rfloor}$ behave like a random walk, we can apply functional central limit theorems (FCLTs) to investigate the behaviour of the limit of $V^{(n)}(t)$. The classical example is Donsker's theorem, which treats the convergence of processes of the form $(1/b_n)(\sum_{k=1}^{\lfloor nt \rfloor} Z_k - a_n t)$ to the Wiener process when the independent random variables $Z_k$ follow a normal distribution. This convergence is with respect to the Skorohod metric on the space of right-continuous functions with existing left limits [36,37]. We refer to such a space of functions on $[0,\infty)$ as $\mathcal{D}([0, \infty), \mathbb{R})$. A key component of the proof of Donsker's theorem is the standard central limit theorem (CLT), but a generalized version of the CLT due to Kolmogorov & Gnedenko [38] also applies to stable distributions with infinite variance and corresponds to a generalized FCLT for $\alpha$-strictly stable processes, called Lévy motions. To describe such Lévy motions, we first need to define a special class of $\alpha$-strictly stable processes.

**Definition 3.9.** For $\alpha \in (0, 2]$ and $\beta \in [-1, 1]$, with $\beta = 0$ when $\alpha = 1$, we define $\alpha$-strictly stable distribution $S(\alpha, \beta)$ to be the distribution with characteristic function

$$\phi(t; \alpha, \beta) = \exp\left(-|t|^\alpha \left(1 - i\beta \operatorname{sgn}(t) \tan\left(\frac{\pi\alpha}{2}\right)\right)\right).$$

With definition 3.9 it can be shown that $\sum_{k=1}^n X_k \sim n^{1/\alpha} S(\alpha, \beta)$ where $X_k \sim S(\alpha, \beta)$ are independent [39]. In fact, this property is usually used to characterize $\alpha$-strictly stable distributions. Note that $S(2, \beta)$ is a normal distribution with mean 0 for any value of $\beta$, while $S(1, 0)$ is a Cauchy distribution.

**Definition 3.10.** A Lévy motion is a stochastic process $V(t; \alpha, \beta)$, $t \geq 0$ with $\alpha \in (0, 2]$ and $\beta \in [-1, 1]$, where $\beta = 0$ when $\alpha = 1$, satisfying the following properties:

(a) Every sample path of $V(t; \alpha, \beta)$ is contained in $D([0, \infty), \mathbb{R})$. Also, $V(0) = 0$.

(b) The increments $V(s_2; \alpha, \beta) - V(s_1; \alpha, \beta), \ldots, V(s_n; \alpha, \beta) - V(s_{n-1}; \alpha, \beta)$ are independent for any $0 \leq s_1 < \ldots < s_n$.

(c) For $s \geq 0$, $t > 0$ we have $V(s + t; \alpha, \beta) - V(s; \alpha, \beta)$ equal in distribution to $t^{1/\alpha} S(\alpha, \beta)$.

A standard example of such a Lévy motion is the Wiener process for $\alpha = 2$. Lévy motions allow the following generalization of Donsker's theorem.

**Theorem 3.11 (FCLT for α-stable Lévy motions).** *Let $Y_1, Y_2, \ldots$ be independent, identically distributed random variables with cumulative distribution $F_Y(x)$ satisfying*

$$1 - F_Y(M) \sim c_+ M^{-\kappa} \text{ as } x \to \infty \quad and \quad F_Y(M) \sim c_- |M|^{-\kappa} \text{ as } x \to -\infty$$

*where $\kappa > 0$, $c_+ \geq 0$ and $c_- \geq 0$ are constants with $c_+$ and $c_-$ not both zero and $c_+ = c_-$ if $\kappa = 1$. Let $\alpha = \min\{\kappa, 2\}$ and $\beta = (c_+ - c_-)/(c_+ + c_-)$. Depending on $\kappa$, choose $a_n$ and $b_n$ from the table below.*

| $\kappa$ | $a_n$ | $b_n$ |
|---|---|---|
| $0 < \kappa < 1$ | $0$ | $\left( \pi(c_+ + c_-) \left( 2\Gamma(\alpha) \sin\left(\frac{\alpha\pi}{2}\right) \right)^{-1} n \right)^{1/\alpha}$ |
| $\kappa = 1$ | $\beta(c_+ + c_-)n\log(n)$ | $\frac{\pi}{2}(c_+ + c_-)n$ |
| $1 < \kappa < 2$ | $n\mathbb{E}[Y_i]$ | $\left( \pi(c_+ + c_-) \left( 2\Gamma(\alpha) \sin\left(\frac{\alpha\pi}{2}\right) \right)^{-1} n \right)^{1/\alpha}$ |
| $\kappa = 2$ | $n\mathbb{E}[Y_i]$ | $(c_+ + c_-)^{1/2}(n\log(n))^{1/2}$ |
| $\kappa > 2$ | $n\mathbb{E}[Y_i]$ | $\left( \mathrm{Var} \frac{(Y_i)}{2} \right)^{1/2} n^{1/2}$ |

*Then the stochastic processes defined by*

$$V^{(n)}(t) = \frac{1}{b_n} \left( \sum_{k=1}^{\lfloor nt \rfloor} Y_k - a_n t \right),$$

*converge, with respect to the Skorohod metric on $D([0, \infty), \mathbb{R})$, to the Lévy motion $V(t; \alpha, \beta)$.*

*Proof.* This is a simple consequence of combining Uchaikin's version of generalized CLT [39] with the discussion of FCLTs arising from CLTs in Whitt [36]. ∎

We can now rephrase this theorem in terms of our sequences of iterates of shift-periodic maps.

**Corollary 3.12.** *Let $F : \mathbb{R} \to \mathbb{R}_\infty$ be a shift-periodic map with integer spikes. Suppose that*

$$\lambda\{y \in [0, 1] : \lfloor F(h(y)) \rfloor > M\} \sim c_+ M^{-\kappa} \quad and \quad \lambda\{y \in [0, 1] : \lfloor F(h(y)) \rfloor < -M\} \sim c_- M^{-\kappa},$$

*as $M \to \infty$, where $\kappa > 0$, $c_+ \geq 0$ and $c_- \geq 0$ are constant with $c_+$ and $c_-$ not both zero and $c_+ = c_-$ if $\kappa = 1$. Choose $\alpha$, $\beta$, $a_n$ and $b_n$ as in theorem 3.11. Let $h$ be as described in theorem 3.3, so that $h(U)$ is an invariant distribution of $F_r$. Define $V^{(n)}(t)$ by (3.6). Then these stochastic processes converge, with respect to the Skorohod metric on $D([0, \infty), \mathbb{R})$, to the Lévy motion $V(t; \alpha, \beta)$.*

Note that while we required $c_+ > 0$ or $c_- > 0$ in corollary 3.12, effectively excluding continuous shift-periodic maps, the random variables $Y_n = \lfloor F^n(h(U)) \rfloor - \lfloor F^{n-1}(h(U)) \rfloor$ generated by a continuous shift-periodic map $F$ with integer spikes have finite second moments, so that Donsker's theorem covers this case and we get Brownian motion upon an appropriate scaling.

Let us briefly return to map $F(x; \kappa)$ in example 2.4. Simple calculations show that the constant $\kappa$ in corollary 3.12 is the same as parameter $\kappa$ of map $F(x; \kappa)$. The corollary then tells us that for appropriately chosen $a_n$ and $b_n$ the stochastic process $V^{(n)}(t)$, as defined in equation (3.6), behaves in a limit like a Lévy motion $Y(t; \alpha, \beta)$. Here $\alpha = \min\{\kappa, 2\}$. In particular, we get a Wiener process for $\kappa \geq 2$. This can also be observed in figure 2.

More general results concerning generalized CLTs and convergence to Lévy motions can be found in the literature, relaxing the independence condition to various mixing conditions. For recent results see,

for example, [40–44]. It is possible to extend statements like in corollary 3.12 to sequences of iterates generated from more general shift-periodic maps, but the conditions from papers [40–44] translate to quite technical restrictions.

In the next section, we look at continuous stochastic processes from a different point of view. Instead of scaling a sequence of iterates by multiplication with a space-scaling constant, we scale the parameters $\varepsilon$ and $\delta$ of map $F(x; \varepsilon, \delta)$ from example 2.3. The resulting process is still confined to $\mathbb{Z}$, but continuous in time, giving dynamical behaviour different from what we have seen in this subsection.

# 4. Continuous-time random walks

In §2, our investigation of shift-periodic maps was motivated by the parameter-dependent map $F(x; \varepsilon, \delta)$ from example 2.3, see figure 1. For most choices of parameters $\varepsilon$ and $\delta$, sequence $\lfloor x_n \rfloor$ cannot have two consecutive jumps, so does not strictly behave like a random walk according to our definition. However, it does display apparent similarities. We will describe in this section in what sense random walk behaviour appears in trajectories generated by this map.

Before we give a precise statement of theorem 4.4, we discuss the invariant density of $F_r(x; \varepsilon, \delta)$. This will be necessary both for motivation and proof of the theorem.

## 4.1. Invariant density

To calculate the invariant density of $F_r(x; \varepsilon, \delta)$, we apply a general result on invariant densities of piecewise linear maps due to Góra [45].

**Lemma 4.1.** *Consider shift-periodic map $F(x; \varepsilon, \delta)$ defined in example 2.3, where $\varepsilon > 0$ and $\delta > 0$. For parameters $A \in [0,1]$ and $B \in \mathbb{R} \setminus \{0\}$ define the indicator function $\mathbb{1} : [0, 1] \to \{0, 1\}$ by*

$$\mathbb{1}(x; A, B) = \begin{cases} 1 & \text{if } x \in [0, A], B > 0 \text{ or } x \in [A, 1], B < 0; \\ 0 & \text{otherwise.} \end{cases}$$

*We define the cumulative derivative $\beta(x, n)$ iteratively by*

$$\beta(x, n) = \beta(x, n - 1) \cdot F_r'(F_r^{n-1}(x; \varepsilon, \delta); \varepsilon, \delta), \quad \text{for } n \geq 2, \quad \text{and} \quad \beta(x, 1) = F_r'(x; \varepsilon, \delta),$$

*where we leave function $\beta(x, n)$ undefined when the derivatives do not exist. This is the case only for finitely many points in $(0, 1)$ for each $n$. We further define two cumulative derivatives at $c_1 = 1/4$ and $c_2 = 3/4$ by $\beta^L(c_i, n) = \lim_{x \to c_i^+} \beta(x, n)$ and $\beta^R(c_i, n) = \lim_{x \to c_i^-} \beta(x, n)$ for $i = 1, 2$. The invariant density of $F(x; \varepsilon, \delta)$ is given by*

$$f_i(x; \varepsilon, \delta) = \frac{1}{K} \left( 1 + \sum_{j=1}^{2} D_j^L \sum_{n=1}^{\infty} \frac{\mathbb{1}(x; F_r^n(c_j), -\beta^L(c_j, n))}{|\beta^L(c_j, n)|} \right.$$
$$\left. + \sum_{j=1}^{2} D_j^R \sum_{n=1}^{\infty} \frac{\mathbb{1}(x; F_r^n(c_j), \beta^R(c_j, n))}{|\beta^R(c_j, n)|} \right), \tag{4.1}$$

*where $K$ is a normalization constant, chosen so that $f_i$ integrates to 1 over $[0,1]$ and $D_1^L, D_2^L, D_1^R, D_2^R$ are constants dependent on $\varepsilon, \delta$ with $D_i^R \to 1$ and $D_i^L \to 1$ as $\varepsilon, \delta \to 0$, for $i \in \{1, 2\}$.*

*Proof.* This is just an application of Góra's results on invariant densities of eventually expanding maps in [45]. Here $D = (D_1^L, D_1^R, D_1^R, D_2^R)$ is the solution of $(-S^T + I)D^T = (1, 1, 1, 1)^T$ where $S = (S_{i,j})$ is a matrix with entries dependent on $F_r^n(c)$ and $\beta(c, n)$, $c \in \{c_1^L, c_1^R, c_2^L, c_2^R\}$, converging to 0 as the parameter $\varepsilon$ and $\delta$ of $F_r^n(x; \varepsilon, \delta)$ converge to 0. For more details on $S$ see p. 7 of [45]. ∎

For $\varepsilon, \delta = 0$ map $F(x; \varepsilon, \delta)$ is linear between grid lines, so of the type discussed in §3, and has invariant density equal 1. So one might expect that $f_i(x; \varepsilon, \delta) \to 1$ as $\varepsilon, \delta \to 0$. This convergence is not uniform on $[0, 1]$, but we can make the important observation below:

**Corollary 4.2.** *Let $f_i(x; \varepsilon, \delta)$ be the invariant density of $F(x; \varepsilon, \delta)$, described in lemma 4.1. For any $d > 0$ we have $f_i(x; \varepsilon, \delta) \to 1$ as $\varepsilon, \delta \to 0$ uniformly on $[d, 1 - d]$.*

*Proof.* Let $d > 0$. For any fixed $n \in \mathbb{N}$, we have that $F_r^n(c_1) \to 0$ and $F_r^n(c_2) \to 1$ as $\varepsilon, \delta \to 0$, so that the length of the interval on which $\mathbb{1}(x; F_r^n(c_j), -\beta^L(c_j, n)) \neq 0$ or $\mathbb{1}(x; F_r^n(c_j), \beta^R(c_j, n)) \neq 0$ also goes to zero,

for $j = 1, 2$. Noting that $|\beta^L(c_j, n)| \geq 2^n$ and $|\beta^R(c_j, n)| \geq 2^n$, then the integral of the weighted sum of four infinite sums in equation (4.1) over [0,1] goes to 0 as $\varepsilon, \delta \to 0$. Adding 1 to this integral, we get $K$. So we have $K \to 1$. By the same argument, for sufficiently small $\varepsilon > 0$ and $\delta > 0$ we have $F_r^n(c_1) < d$ and $F_r^n(c_2) > 1 - d$ for all $n \in \{1, 2, \ldots, N\}$, so that we achieve bound

$$\frac{1}{K} \leq f_i(x; \varepsilon, \delta) \leq \frac{1}{K}\left(1 + \frac{D_1^L + D_2^L + D_1^R + D_2^R}{2^N}\right)$$

valid on $[d, 1-d]$ for small $\varepsilon$ and $\delta$. But $N$ was chosen arbitrarily. Recall from lemma 4.1 that also $D_i^R, D_i^L \to 1$ for $i = 1, 2$. Combining these results, we find that $f_i(x; \varepsilon, \delta) \to 1$ uniformly on $[d, 1-d]$. ∎

## 4.2. From maps to continuous-time random walks

Now consider $X$ distributed according to the invariant distribution of $F_r(x; \varepsilon, \delta)$ on [0,1] and $X_n = \lfloor F^n(X; \varepsilon, \delta)\rfloor$. For now we say a jump occurs when $X_n \neq X_{n+1}$. By corollary 4.2 the invariant density of $F_r(x; \varepsilon, \delta)$ is close to 1 at the spikes 1/4 and 3/4 of the map for small parameters $\varepsilon$ and $\delta$. Further, direct calculation shows that the length of the subset of [0,1] mapped outside the unit interval by $F(x; \varepsilon, \delta)$ is

$$\ell(\varepsilon) + \ell(\delta) \quad \text{where} \quad \ell(x) = \frac{x(3+x)}{2(x+2)(x+4)}. \tag{4.2}$$

This calculation suggests that the probability of a jump is about $3(\varepsilon + \delta)/16$. However, successive jump probabilities are not independent, since for small parameters successive jumps are impossible. We fix this issue by introducing a scaling in time, together with a scaling of the parameters. This scaling gives us behaviour resembling a continuous-time random walk, defined below.

**Definition 4.3.** Consider a continuous-time stochastic process $Y(t)$, $t \geq 0$ with $Y(0) = 0$ which takes values in $\mathbb{Z}$ and is right-continuous. Let $T_0 = 0$. For $j \geq 1$ define the time of the $j$-th jump by

$$T_j = \min\{t \in (T_{j-1}, \infty): Y(t) \neq Y(T_{j-1})\}.$$

Suppose $T_1, T_2, \ldots$ are independent. Then we say $Y(t)$ is a continuous-time random walk.

**Theorem 4.4.** Let $\delta, \varepsilon > 0$. For $m \in \mathbb{N}$ let $X_m$ be distributed according to the invariant distribution, $f_i(x; \varepsilon/m; \delta/m)$, with respect to $F(x; \varepsilon/m, \delta/m)$. Define

$$Y_m(t) = \left\lfloor F^{\lfloor mt\rfloor}\left(X_m; \frac{\varepsilon}{m}, \frac{\delta}{m}\right)\right\rfloor.$$

Let $T_{m,1}, T_{m,2}, \ldots$ denote the jump times of $Y_m(t)$, that is,

$$T_{m,j} = \min\{t \in (T_{m,j-1}, \infty): Y_m(t) \neq Y_m(T_{m,j-1})\},$$

where $T_{m,0} = 0$. Then for any $k \geq 0$ we have

$$\mathbb{P}\left(T_{m,k+1} \leq \frac{\lfloor mt_k\rfloor}{m} + \tau \ \bigg| \ T_{m,k} = \frac{\lfloor mt_k\rfloor}{m}, \ \ldots, \ T_{m,1} = \frac{\lfloor mt_1\rfloor}{m}\right) \to 1 - \exp(-\gamma\tau)$$

as $m \to \infty$, where

$$\gamma = \frac{3(\delta + \varepsilon)}{16}. \tag{4.3}$$

Let $Y(t)$ be a continuous-time random walk with waiting times, $T_j - T_{j-1}$, exponentially distributed with mean $1/\gamma$. Then theorem 4.4 says that for $Y_m(t)$ the probability of the $(k+1)$-th jump occurring at time $\lfloor mt_k\rfloor/m + \tau$, given that the first $k$ jumps occurred at times $\lfloor mt_1\rfloor/m, \lfloor mt_2\rfloor/m, \ldots, \lfloor mt_k\rfloor/m$, converges to the probability of the $(k+1)$-th jump of $Y(t)$ occurring at time $t_k + \tau$, given that the first $k$ jumps occurred at $t_1, t_2, \ldots, t_k$.

## 4.3. Conditionally invariant distribution

While we discussed invariant distributions of $F_r(x; \varepsilon, \delta)$ above, to prove theorem 4.4, it will often be more convenient to work with conditional distributions on [0,1] which are invariant with respect to $F(x; \varepsilon, \delta)$ conditioned on the event that the iterative sequence stays in [0,1].

The use of conditionally invariant densities is common in investigations of dynamical systems with holes, that is, trajectories generated by maps $F : \Omega \to F(\Omega)$ with $\Omega \subsetneq F(\Omega)$ until the point of escape from $\Omega$. Early results are due to Pianigiani & Yorke [46], who motivated the discussion with the example of a billiard table with chaotic trajectories, and introduced the concept of conditionally invariant measures. This idea has been investigated further by other authors, for example, Demers and Young studied escape rates through the small holes in [47].

In our case, it will be useful to work from the point of view of a dynamical system with holes whenever no jump is occurring. For a probability density $f$ of a distribution on [0,1], the density after application of $F(x; \varepsilon, \delta)$, conditional on not mapping outside [0,1], is given by the Frobenius–Perron operator [46]

$$\mathcal{P}_{\varepsilon,\delta}(f)(t) = \begin{cases} \frac{1}{C}\left( \frac{f(F_1^{-1}(t))}{4+\varepsilon} + \frac{f(F_3^{-1}(t))}{2+\delta} + \frac{f(F_4^{-1}(t))}{4+\delta} \right) & \text{if } t \in \left(0, \frac{1}{2}\right), \\ \frac{1}{C}\left( \frac{f(F_1^{-1}(t))}{4+\varepsilon} + \frac{f(F_2^{-1}(t))}{2+\varepsilon} + \frac{f(F_4^{-1}(t))}{4+\delta} \right) & \text{if } t \in \left(\frac{1}{2}, 1\right), \end{cases} \tag{4.4}$$

where $F_1^{-1}$, $F_2^{-1}$, $F_3^{-1}$, $F_4^{-1}$ denote the inverses of $F(x; \varepsilon, \delta)$ restricted to (0, 1/4), (1/4, 1/2), (1/2, 3/4), (3/4, 1), respectively, and normalization constant $C$ is chosen so that $\mathcal{P}_{\varepsilon,\delta}$ integrates to 1 over the unit interval.

**Lemma 4.5.** *There exists a unique density $f_c(x; \varepsilon, \delta)$ with $f_c(x; \varepsilon, \delta) = v$ on $(0, 1/2)$ and $f_c(x; \varepsilon, \delta) = 2 - v$ on $(1/2, 1)$ such that $\mathcal{P}_{\varepsilon,\delta}(f_c) = f_c$. We will subsequently call this the conditionally invariant density. It satisfies $f_c(x; \varepsilon, \delta) \to 1$ as $\varepsilon, \delta \to 0$.*

*Proof.* Suppose $f$ is a density which is constant equal $v$ on (0, 1/2) and $1 - v$ on (1/2, 1). It satisfies $\mathcal{P}_{\varepsilon,\delta}(f) = f$ if and only if $v$ satisfies the following equation:

$$v\left( \frac{1}{2}\left( \frac{v}{4+\varepsilon} + \frac{2-x}{2+\delta} + \frac{2-v}{4+\delta} \right) + \frac{1}{2}\left( \frac{v}{4+\varepsilon} + \frac{v}{2+\varepsilon} + \frac{2-v}{4+\delta} \right) \right) = \left( \frac{v}{4+\varepsilon} + \frac{2-v}{2+\delta} + \frac{2-v}{4+\delta} \right).$$

This equation is obtained from the first line of equation (4.4), the left corresponds to normalization constant $C$ multiplied by $v$. Solving this quadratic equation, we obtain for all $\varepsilon, \delta \geq 0$ a unique solution $v$ with both $v \geq 0$ and $2 - v \geq 0$, and also the unique conditionally invariant density $f(x; \varepsilon, \delta)$ described in lemma 4.5. This solution linearizes to

$$f_c(x; \varepsilon, \delta) \approx \begin{cases} 1 + \frac{\varepsilon}{12} - \frac{\delta}{12} & \text{if } x \in \left(0, \frac{1}{2}\right), \\ 1 - \frac{\varepsilon}{12} + \frac{\delta}{12} & \text{if } x \in \left(\frac{1}{2}, 1\right), \end{cases}$$

for small $\varepsilon, \delta$. In particular, $f_c(x; \varepsilon, \delta) \to 1$ as $\varepsilon, \delta \to 0$. ∎

## 4.4. Convergence to the conditionally invariant density

In this subsection, we make a first step towards proving theorem 4.4. We show for some initial densities $k$ that $\mathcal{P}_{\varepsilon,\delta}^n(k)$ does not only converge to the corresponding conditionally invariant density $f_c$, but that there is an upper bound on the convergence speed which works for all $\varepsilon > 0$ and $\delta > 0$.

Pianigiani and Yorke extensively studied the existence of and convergence to conditionally invariant densities for expanding maps in [46]. While their approach does not give us the desired bound on convergence speed, one of their results, the lemma stated below, will be very useful in our proof.

**Lemma 4.6 (Pianigiani–Yorke).** *Let $\mathcal{P}_F$ be the Frobenius–Perron operator corresponding to a map $F$ on [0, 1]. Suppose $f$, $g$ are Lebesgue integrable over [0,1] with $\int_0^1 f(x)\,dx = \int_0^1 g(x)\,dx = 1$, $\inf_{[0,1]} f(x) > 0$, $\sup_n \|\mathcal{P}_F^n(g)\|_\infty < \infty$ and $\sup_n \|\mathcal{P}_F^n(1)\|_\infty < \infty$. Then there exists some $L$ such that for $n \geq 1$*

$$\|\mathcal{P}_F^n(f) - \mathcal{P}_F^n(g)\|_\infty \leq L\|f - g\|_\infty$$

*is satisfied. More precisely, we may take*

$$L = \frac{1}{\inf_{[0,1]} f}\left( \sup_n \|\mathcal{P}_F^n(1)\|_\infty + \sup_n \|\mathcal{P}_F^n(g)\|_\infty \right).$$

*Proof.* We can apply [46, proposition 1]. The original statement requires that $f$, $g \in K$ where $K = \{f \in C([0, 1]) : \sup_{[0,1]} f(x) < \infty, \inf_{[0,1]} f(x) \mathrm{d}x > 0, \int_0^1 f(x) \mathrm{d}x = 1\}$, but the proof also works for the assumptions of lemma 4.6. Note that in this case, we define the infinum by $\inf_{[0,1]} f = \inf \{s \in \mathbb{R} : \lambda(\{x \in [0, 1] : f(x) < s\}) = 0\}$. ∎

We now have all the necessary tools to prove theorem 4.4. The key idea is to approximate densities by piecewise constant densities.

**Lemma 4.7.** *Let* $\mu > 0$. *Then there exists* $\omega > 0$ *and* $N \in \mathbb{N}$ *such that for any piecewise constant density* $k : [0,1] \to [0,2]$ *with*

$$k(t) = \begin{cases} x & \text{if } t \in \left(0, \frac{1}{2}\right), \\ 2 - x & \text{if } t \in \left(\frac{1}{2}, 1\right), \end{cases} \tag{4.5}$$

*whenever* $n \geq N$, $x \in [0,2]$ *and* $0 < \varepsilon$, $\delta < \omega$, *then* $\|\mathcal{P}^n_{\varepsilon,\delta}(k) - f_c(\,\cdot\,;\varepsilon,\delta)\|_\infty < \mu$, *where the map* $f_c(\,\cdot\,;\varepsilon,\delta) : [0, 1] \to \mathbb{R}$ *is the conditionally invariant density.*

*Proof.* By [46, theorem 3], densities $\mathcal{P}^n_{\varepsilon,\delta}(k)$ converge to invariant density $f_c$ as $n \to \infty$. Lemma 4.7 will now show that this convergence is uniform over all choices of $k$. So let $k$ be an arbitrary function satisfying the conditions of the Lemma. Density $\mathcal{P}^n_{\varepsilon,\delta}(k)$ is constant for each $n \in \mathbb{N}$ on both $(0, 1/2)$ and $(1/2, 1)$. First, we bound ratio

$$r(x, \varepsilon, \delta) = \frac{k - \mathcal{P}_{\varepsilon,\delta}(k)}{\mathcal{P}_{\varepsilon,\delta}(k) - \mathcal{P}^2_{\varepsilon,\delta}(k)},$$

where $x$ is the value of $k$ appearing in equation (4.5). Note that on $(0, 1/2)$

$$\mathcal{P}_{\varepsilon,\delta}(k) = \frac{1}{c_1(x)} \left( \frac{x}{4 + \varepsilon} + \frac{2 - x}{2 + \delta} + \frac{2 - x}{4 + \delta} \right) = \frac{a_1(x)}{c_1(x)};$$

where $c_1(x)$ is the normalization constant $C$ from formula (4.4). Moreover, we obtain $\mathcal{P}_{\varepsilon,\delta}(k) = 2 - a_1(x)/c_1(x)$ on $(1/2, 1)$ from normalization. Furthermore, on $(0, 1/2)$,

$$\mathcal{P}^2_{\varepsilon,\delta}(k) = \frac{1}{c_2(x)} \left( \frac{a_1(x)/c_1(x)}{4 + \varepsilon} + \frac{2 - a_1(x)/c_1(x)}{2 + \delta} + \frac{2 - a_1(x)/c_1(x)}{4 + \delta} \right) = \frac{a_2(x)}{c_1(x)c_2(x)},$$

where $c_2(x)$ is again the normalization constant $C$ from formula (4.4) and $a_2(x)$ is a linear polynomial equal to $c_1(x)c_2(x)\mathcal{P}^2_{\varepsilon,\delta}(k)$. Note that for fixed parameters $\varepsilon$ and $\delta$ denominators $c_1(x)$ and $c_1(x)c_2(x)$ can also be written as linear polynomials of $x$ and are non-zero. With this notation $r$ can be expressed as

$$r(x, \varepsilon, \delta) = \frac{c_1^2(x)c_2(x)x - c_1(x)c_2(x)a_1(x)}{c_1(x)c_2(x)a_1(x) - c_1(x)a_2(x)},$$

a quotient of two polynomials. As a quotient of a cubic and quadratic polynomial, an explicit calculation shows that denominator and enumerator have the same positive root and that this root has multiplicity 1 and is equal to the value of the conditionally invariant density $v$ from lemma 4.5. So $r$ can be extended to a continuous function in $x$, $\varepsilon$ and $\delta$ for $x \geq 0$, $\varepsilon \geq 0$, $\delta \geq 0$. For $\varepsilon = \delta = 0$ a calculation gives $r(x, 0, 0) = -2$. By continuity we can choose some $\omega > 0$ such that $\varepsilon$, $\delta < \omega$ and $x \in [0,2]$ implies $|r(x, \varepsilon, \delta)| > 3/2$. By repeatedly applying this result,

$$|\mathcal{P}^n_{\varepsilon,\delta}(k)(y) - \mathcal{P}^{n+1}_{\varepsilon,\delta}(k)(y)| < \left(\frac{2}{3}\right)^n |k(y) - \mathcal{P}_{\varepsilon,\delta}(k)(y)| \leq 2\left(\frac{2}{3}\right)^n, \quad \text{for } y \in \left(0, \frac{1}{2}\right) \cup \left(\frac{1}{2}, 1\right)$$

But as each $\mathcal{P}^n_{\varepsilon,\delta}(k)$ is constant on $(0, 1/2)$ and equal to $2 - \mathcal{P}^n_{\varepsilon,\delta}(k)(1/4)$ on $(1/2-1)$, it follows that $\|\mathcal{P}^n_{\varepsilon,\delta}(k) - f_c(\,\cdot\,;\varepsilon,\delta)\|_\infty \leq 6(2/3)^n$ for each $n$. ∎

Densities with three constant pieces are more convenient for approximating a density conditional on a jump having just occurred, something we will look at in the later parts of the proof of theorem 4.4. So we now focus our attention on such densities.

**Definition 4.8.** For $S \geq 0$, we define set $K_S$ of piecewise constant densities $k : [0,1] \to [0,\infty)$ with $\int_0^1 k(t) \, \mathrm{d}t = 1$, which can be written in one of the following two forms

$$k(t) = \begin{cases} k_1 & \text{if } t \in (0, b), \\ k_2 & \text{if } t \in \left(b, \frac{1}{2}\right), \\ k_3 & \text{if } t \in \left(\frac{1}{2}, 1\right), \end{cases} \quad \text{where } |k_1 - k_2| \leq S \text{ for } b \in \left(0, \frac{1}{2}\right), \tag{4.6}$$

or

$$k(t) = \begin{cases} k_1 & \text{if } t \in \left(0, \frac{1}{2}\right), \\ k_2 & \text{if } t \in \left(\frac{1}{2}, b\right), \quad \text{where } |k_2 - k_3| \leq S \text{ for } b \in \left(\frac{1}{2}, 1\right), \\ k_3 & \text{if } t \in (b, 1), \end{cases} \tag{4.7}$$

where $k_1$, $k_2$, $k_3$ are non-negative constants. We define further map $\Psi : K_S \to [0, \infty)$ with

$$\Psi(k) = \begin{cases} |k_1 - k_2| & \text{for } b \in \left(0, \frac{1}{2}\right) \\ |k_2 - k_3| & \text{for } b \in \left(\frac{1}{2}, 1\right), \end{cases} \tag{4.8}$$

where $b$ is as defined in equations (4.6)–(4.7).

**Lemma 4.9.** *Let $S > 0$. Then there exist $\omega > 0$ and $B \in [0, 1)$ such that for $0 < \varepsilon$, $\delta < \omega$ and any $k \in K_S$ we have $\mathcal{P}_{\varepsilon,\delta}(k) = k' \in K_S$ with $\Psi(k') \leq B\, \Psi(k)$.*

*Proof.* Let $k \in K_S$. Let $b$ be as defined in equations (4.6)–(4.7). First, assume $b(4 + \varepsilon) < 1/2$. A direct calculation leads to

$$\mathcal{P}_{\varepsilon,\delta}(k)(t) = \begin{cases} k_1' = \frac{1}{C}\left(\frac{k_1}{4+\varepsilon} + \frac{k_3}{2+\delta} + \frac{k_3}{4+\delta}\right) & \text{if } t \in (0, b(4+\varepsilon)), \\ k_2' = \frac{1}{C}\left(\frac{k_2}{4+\varepsilon} + \frac{k_3}{2+\delta} + \frac{k_3}{4+\delta}\right) & \text{if } t \in \left(b(4+\varepsilon), \frac{1}{2}\right), \\ k_3' = \frac{1}{C}\left(\frac{k_2}{4+\varepsilon} + \frac{k_2}{2+\varepsilon} + \frac{k_3}{4+\delta}\right) & \text{if } t \in \left(\frac{1}{2}, 1\right), \end{cases}$$

where $C$ is the normalization constant. The difference between the values of $\mathcal{P}_{\varepsilon,\delta}(k)$ on $(0, 1/2)$ is given by $|k_1' - k_2'| = |k_1 - k_2| / (C(4 + \varepsilon))$. Proceeding in the same way for all other possible choices of $b$ we get

$$\Psi(k') \leq \frac{\Psi(k)}{(2C)}. \tag{4.9}$$

Now we want to bound $C$ below. Say $b > 1/2$. If $k_2 > m$, then $k_3 > m - S$ and as the density is non-negative, $\int_0^1 k(t)\, dt > (m - S)/2$. But as $\int_0^1 k(t)\, dt = 1$ we get a contradiction for $m \geq 2 + S$. Similarly, if instead $k_3 > m$. We also need $k_1 \leq 2$. For $k$ constant on $(1/2, 1)$, we proceed in the same way. So all functions in $K_S$ are bounded above by $2 + S$. Let $\ell(\varepsilon) + \ell(\delta)$ be the Lebesgue measure of the subset of $[0, 1]$ which is mapped outside the unit interval by $F(t; \varepsilon, \delta)$, as in equation (4.2). Then $\ell(\varepsilon) + \ell(\delta) \to 0$ as $\varepsilon$, $\delta \to 0$. For sufficiently small parameters, say $0 < \varepsilon$, $\delta < \omega$, we will have normalization constant $C \geq 1 - (\ell(\varepsilon) + \ell(\delta))(2 + S) \geq 2/3$. Then substituting into inequality (4.9), we obtain inequality (4.9). ∎

Next, we note how lemma 4.6 can be applied to densities of this piecewise constant form.

**Corollary 4.10.** *Let $S \geq 0$ and $s > 0$. Let $K_S$ be as described in lemma 4.9. There exists $L > 0$ and $\omega > 0$ such that for any $0 < \varepsilon$, $\delta < \omega$, $g \in K_S$, density $f$ with $\inf_{[0,1]} f > s$ and $n \geq 1$*

$$\left\| \mathcal{P}_{\varepsilon,\delta}^n(f) - \mathcal{P}_{\varepsilon,\delta}^n(g) \right\|_\infty \leq L \|f - g\|_\infty.$$

*Proof.* By lemma 4.9 there exists $\omega > 0$ such that $0 < \varepsilon$, $\delta < \omega$ and $g \in K_S$ imply $\mathcal{P}_{\varepsilon,\delta}^n(g) \in K_S$ for each $n \geq 1$. Using the last paragraph of the proof of lemma 4.9, then $\sup_n \|\mathcal{P}_{\varepsilon,\delta}^n(g)\|_\infty \leq 2 + S$. Also, since $\mathcal{P}_{\varepsilon,\delta}(1)$ is constant on both $(0, 1/2)$ and $(1/2, 1)$, we have $\|\mathcal{P}_{\varepsilon,\delta}^n(1)\|_\infty \leq 2$. By applying lemma 4.6, we find that for $0 < \varepsilon$, $\delta < \omega$, density $f$ with $\inf_{[0,1]} f > s$ satisfies

$$\left\| \mathcal{P}_{\varepsilon,\delta}^n(f) - \mathcal{P}_{\varepsilon,\delta}^n(g) \right\|_\infty \leq L \|f - g\|_\infty,$$

where $L = (4 + S)/s$. ∎

**Lemma 4.11.** *Let $\mu > 0$ and $S > 0$. There exist $\omega > 0$ and $N \in \mathbb{N}$ such that for all $n \geq N$, $0 < \varepsilon$, $\delta < \omega$ and $k \in K_S$ we have $\|\mathcal{P}_{\varepsilon,\delta}^n(k) - f_{c_{\varepsilon,\delta}}\|_\infty < \mu$.*

*Proof.* First, use lemma 4.9 to observe that for any $S_1 \in (0, S)$ there exist $\omega_1 > 0$ and $N_1 \in \mathbb{N}$ such that for any $k \in K_S$ and $0 < \varepsilon$, $\delta < \omega_1$ we have $\mathcal{P}_{\varepsilon,\delta}^{N_1}(k) \in K_{S_1}$. But $S_1$ can be chosen small enough that there is some $s > 0$ such that for any $k \in K_S$ and $N_2 \geq N_1 + 1$ we have $\inf_{[0,1]} \mathcal{P}_{\varepsilon,\delta}^{N_2}(k) > s$. Apply corollary 4.10 with $f = \mathcal{P}_{\varepsilon,\delta}^{N_2}(k)$ to find some $L > 0$ and $\omega_2 > 0$ such that when $0 < \varepsilon$, $\delta < \omega_2$, $N_2 \geq N_1 + 1$ and $g \in K_S$ then

$$\|\mathcal{P}_{\varepsilon,\delta}^{N_2 + n}(k) - \mathcal{P}_{\varepsilon,\delta}^n(g)\|_\infty \leq L \|\mathcal{P}_{\varepsilon,\delta}^{N_2}(k) - g\|_\infty. \tag{4.10}$$

Also by lemma 4.9, we can choose $N_2$ such that for any $k \in K_S$ and $0 < \varepsilon,\ \delta < \min\{\omega_1,\ \omega_2\}$, there exists a piecewise constant density $g \in K_S$ such that $g$ is constant on $(0,\ 1/2)$, $g$ is constant on $(1/2,\ 1)$ and $\|\mathcal{P}^{N_2}_{\varepsilon,\delta}(k) - g\|_\infty < \mu/(2L)$. Then by equation (4.10), we get

$$\|\mathcal{P}^{N_2+n}_{\varepsilon,\delta}(k) - \mathcal{P}^n_{\varepsilon,\delta}(g)\|_\infty \leq \frac{\mu}{2}$$

for $n \geq 1$. By lemma 4.7 we also find $N_3 \in \mathbb{N}$ and $\omega_3 > 0$ such that for $0 < \varepsilon,\ \delta < \omega_3$ and $n \geq N_3$

$$\|\mathcal{P}^n_{\varepsilon,\delta}(g) - f_{c_{\varepsilon,\delta}}\| < \frac{\mu}{2}.$$

Now take $N = N_2 + N_3$ and $\omega = \min\{\omega_1,\ \omega_2,\ \omega_3\}$ to complete the proof. ∎

## 4.5. Proof of theorem 4.4

Recall from corollary 4.2 that invariant density $f_{i,m}(x)$ converges uniformly to 1 on $[d,\ 1-d]$ as $m \to \infty$ for any $d > 0$. In our proof, we need this convergence to be extended to all of $[0,1]$. Therefore, we first prove an alternate version of theorem 4.4, in which we do not start with the invariant density $f_i$, but a related density $f'_i$ and process $Y'_m$, defined below.

**Definition 4.12.** Fix some $0 < d < 1/2$ and define

$$f'_i\left(x;\ \frac{\varepsilon}{m},\ \frac{\delta}{m}\right) = \begin{cases} f_i\left(x;\ \frac{\varepsilon}{m},\ \frac{\delta}{m}\right) & \text{if } x \in [d,\ 1-d] \\ k_m & \text{elsewhere,} \end{cases} \tag{4.11}$$

where $k_m$ is chosen so that $f'_i(x,\ \varepsilon/m,\ \delta/m)$ integrates to 1 over $[0,1]$. Define

$$Y'_m(t) = \left\lfloor F^{\lfloor mt \rfloor}\left(X'_m;\ \frac{\varepsilon}{m},\ \frac{\delta}{m}\right) \right\rfloor,$$

where $X'_m$ is distributed according to $f'_i(x;\ \varepsilon/m,\ \delta/m)$. Let $T'_{m,1},\ T'_{m,2},\ \ldots$ denote the jump times of $Y'_m(t)$, that is, $T'_{m,j} = \min\{t \in [T'_{m,j-1},\ \infty) : Y'_m(t) \neq Y'_m(T'_{m,j-1})\}$ where $T'_{m,0} = 0$.

We now describe the behaviour of the first jump $T'_{m,1}$ for process $Y'_m(t)$. To simplify notation, set

$$f_{c,m}(x) = f_c\left(x;\ \frac{\varepsilon}{m},\ \frac{\delta}{m}\right), \quad F_m(x) = F\left(x;\ \frac{\varepsilon}{m},\ \frac{\delta}{m}\right),$$

$$f_{i,m}(x) = f_i\left(x;\ \frac{\varepsilon}{m},\ \frac{\delta}{m}\right), \quad \mathcal{P}_m = \mathcal{P}_{\varepsilon/m,\delta/m},$$

$$f'_{i,m}(x) = f'_i\left(x;\ \frac{\varepsilon}{m},\ \frac{\delta}{m}\right).$$

**Lemma 4.13.** Let $\delta,\ \varepsilon > 0$. Let $0 < 1/2 < d$. Let $Y'_m(t)$ and $T'_{m,1}$ be as described in definition 4.11. Then for $\tau > 0$ we have $\mathbb{P}(T'_{m,1} \leq \tau) \to 1 - \exp(-\gamma\tau)$ as $m \to \infty$, where $\gamma$ is given by equation (4.3).

*Proof.* Apply corollary 4.10 with $s = 1/2$, $S = 0$ to find $L > 0$ such that for large enough $m$, say $m \geq M_1$, and densities $f$ with $\inf_{[0,1]} f > 1/2$

$$\|\mathcal{P}^n_m(f) - \mathcal{P}^n_m(1)\|_\infty \leq L\|f - 1\|_\infty. \tag{4.12}$$

Choose $\mu > 0$ small enough so that $\mu(1 + 2d) < d$. Then for large enough $m$, say $m \geq M_2 \geq M_1$, we have $|f_{i,m}(x) - 1| < \mu$ on $[d,\ 1-d]$ and by considering bounds on $k_m$, the value of $f'_{i,m}$ on $(0,\ d) \cup (1-d,\ 1)$, we get $\|f'_{i,m} - 1\|_\infty \leq \mu(1 + 1/(2d))$. Since $\mu(1 + 1/(2d)) < 1/2$, we have $f'_{i,m}$ bounded below by $1/2$. Then using equation (4.12) we have

$$\|\mathcal{P}^n_m(f'_{i,m}) - \mathcal{P}^n_m(1)\|_\infty \leq L\mu\left(1 + \frac{1}{2d}\right), \tag{4.13}$$

when $n \geq 1$ and $m \geq M_2$. By lemma 4.7, there also exists some $M_3 \geq M_2$ and $N \in \mathbb{N}$ such that when $m \geq M_3$ and $n \geq N$ we have $\|\mathcal{P}^n_m(1) - f_{c,m}\|_\infty < \mu$. Write $B(d) = 1 + L(1 + 1/(2d))$ and observe, using equation (4.13), that whenever $m \geq M_3$ and $n \geq N$

$$\|\mathcal{P}^n_m(f'_{i,m}) - f_{c,m}\|_\infty \leq \mu B(d). \tag{4.14}$$

Let $\tau > 0$. Then pick $m > \max\{M_3, N/\tau\}$ and consider the probability that no jump occurs until time $\tau$ for $Y'_m(t)$,

$$\mathbb{P}(T'_{m,1} > \tau) = \prod_{n=1}^{\lfloor m\tau \rfloor} \mathbb{P}(F^n_m(X'_m) \in [0, 1) \mid F^1_m(X'_m), \ldots, F^{n-1}_m(X'_m) \in [0, 1)).$$

Write $A_m$ for the subset of [0,1] mapped outside the unit interval by $F_m$. As in equation (4.2), denote by $\ell(\varepsilon/m) + \ell(\delta/m)$ the length of $A_m$. Write $f_{c,m} = \nu_m$ on $(0, 1/2)$ and $f_{c,m} = 2 - \nu_m$ on $(1/2, 2)$. Using equation (4.14), we get the following upper estimate

$$\prod_{n=N+1}^{\lfloor m\tau \rfloor} \mathbb{P}(F^n_m(X'_m) \in [0, 1) | F^1_m(X'_m), \ldots, F^{n-1}_m(X'_m) \in [0, 1))$$

$$= \prod_{n=N+1}^{\lfloor m\tau \rfloor} \int_{[0,1] \setminus A_m} \mathcal{P}^{n-1}_m (f'_{i,m})(x) \, dx$$

$$\leq \left( 1 - (\nu_m - \mu B(d)) \ell\left(\frac{\varepsilon}{m}\right) - (2 - \nu_m - \mu B(d)) \ell\left(\frac{\delta}{m}\right) \right)^{\lfloor m\tau \rfloor - N}.$$

Since $m \ell(\varepsilon/m) + m \ell(\delta/m) \to \gamma$ as $m \to \infty$, where $\gamma$ is given by equation (4.3), and $\nu_m \to 1$ as $m \to \infty$ by lemma 4.5, the upper bound converges to $\exp[-\gamma(1 - \mu B(d))\tau]$ as $m \to \infty$. By a similar argument, we have a lower bound converging to $\exp[-\gamma(1 + \mu B(d))\tau]$ as $m \to \infty$.

Let $A^N_m$ be the subset of [0,1], which is mapped outside [0,1] within at most $N$ applications of $F_m$. The Lebesgue measure of $A^N_m$ goes to 0 as $m \to \infty$. From equation (4.1), we note that $\sup_m \|f_{i,m}\|_\infty < \infty$ and also $\sup_m \|f'_{i,m}\|_\infty < \infty$. So by integrating over $A^N_m$, we obtain

$$\prod_{n=1}^N \mathbb{P}(F^n_m(X'_m) \in [0, 1) \mid F^1_m(X'_m), \ldots, F^{n-1}_m(X'_m) \in [0, 1)) = 1 - \int_{A^N_m} f'_{i,m}(x) \, dx \to 1$$

as $m \to \infty$. So $\mathbb{P}(T'_{m,1} > \tau)$ is bounded above by a product converging to $\exp[-\gamma(1 - \mu B(d))\tau]$ and below by a product converging to $\exp[-\gamma(1 + \mu B(d))\tau]$ as $m \to \infty$. But $\mu > 0$ was arbitrary, so $\mathbb{P}(T'_{m,1} \leq \tau) \to 1 - \exp(-\gamma\tau)$ as $m \to \infty$. ∎

Now we will prove an alternate version of theorem 4.4 for $Y'_m(t)$. Lemma 4.13 established such a statement already for the first jump. The key component in the general proof will be the following lemma, which helps in describing how the densities develop after a jump, conditional on no further jump occurring, provided we start off close to the conditionally invariant density $f_{c,m}$.

**Lemma 4.14.** *Let*

$$A^1_m = \left( \frac{1}{4} - \frac{\varepsilon/m}{4(4 + \varepsilon/m)}, \frac{1}{4} \right)$$

*denote the subset of (0, 1/4) mapped outside of [0,1] by $F_m$. Take $0 < \mu < 1/4$ and let $g_m$ be a density with $\|g_m - f_{c,m}\|_\infty < \mu$. Let $V_m$ be distributed according to that density and $g^\star$ denote the density corresponding to the distribution of $F_r(V_m; \varepsilon/m, \delta/m)$, conditional on $V_m \in A^1_m$. Then there exists $B > 0$, $M \in \mathbb{N}$ and $N \in \mathbb{N}$ such that for all $m \geq M$ and $n \geq N + \lceil \log(m/\varepsilon)/\log(4 + \varepsilon/m) \rceil$ and for any sequence of $g_m$ satisfying above properties we have*

$$\|\mathcal{P}^n_m(g^\star_m) - f_{c,m}\|_\infty < \mu B. \tag{4.15}$$

*Proof.* Since $V_m \in A^1_m$, we have $F_r(V_m; \varepsilon/m, \delta/m) \in [0, \varepsilon/(4m)]$. On $(\varepsilon/(4m), 1)$ we then have $g^\star_m = 0$, while we have

$$g^\star_m(x) = \frac{1}{c_0} \left( g_m \left( \frac{1+x}{4 + \varepsilon/m} \right) \right), \quad \text{for } x \in \left( 0, \frac{\varepsilon}{4m} \right), \tag{4.16}$$

where $c_0$ is a constant so that $g^\star_m$ integrates to 1 over [0,1]. Let $u_m$ denote the smallest integer such that $F^{u_m+1}_r(1/4; \varepsilon/m, \delta/m) = (4 + \varepsilon/m)^{u_m} \varepsilon/(4m) \geq 1/4$. Then $u_m = \lceil \log(m/\varepsilon)/\log(4 + \varepsilon/m) \rceil$. From equation (4.4), we deduce that for $n \leq u_m$, density $\mathcal{P}^n_m(g^\star_m)$ is obtained from $g^\star_m$ via a scaling of the form

$$\mathcal{P}^n_m(g^\star_m)(x) = \frac{1}{c_n} g^\star_m \left( \frac{x}{(4 + \varepsilon/m)^n} \right) = \frac{1}{c_0 c_n} \left( g_m \left( \frac{1 + x/(4 + \varepsilon/m)^n}{4 + \varepsilon/m} \right) \right), \tag{4.17}$$

where $c_n$ is a constant dependent on $m$, such that $\mathcal{P}_m^n(g_m^\star)(x)$ integrates to 1 over [0,1]. By lemma 4.5, there exists $M_1 \in \mathbb{N}$ such that $m \geq M_1$ implies $1/2 < f_{c,m} < 3/2$. Since $\mu < 1/4$, we obtain a bound $g_m(y) \geq 1/4$ and estimate

$$\mathcal{P}_m^{u_m}(g_m^\star)(x) \geq \frac{1}{4c_0 c_{u_m}} \quad \text{for } x \in I_{u_m} = \left(0, \frac{\varepsilon}{4m}\left(4 + \frac{\varepsilon}{m}\right)^{u_m}\right). \tag{4.18}$$

Using $(4 + \varepsilon/m)^{u_m}\varepsilon/(4m) \geq 1/4$ and equation (4.18), we get $c_0 c_{u_m} \geq 1/16$, since $\mathcal{P}_m^{u_m}(g_m^\star)$ must integrate to 1 over [0,1]. But recall that $\|g_m - f_{c,m}\|_\infty < \mu$ and $f_{c,m}$ constant on $(0, 1/2)$. Combining this with equation (4.17) gives us that the values of $\mathcal{P}_m^{u_m}(g_m^\star)$ on $I_{u_m}$ are contained in a subinterval of $(0, \infty)$ of length $32\mu$. But applying equation (4.4) again, we see that for $\mathcal{P}_m^{u_m+1}(g_m^\star)$ the unit interval can be split into three subintervals $(0, b_1)$, $(b_1, b_2)$ and $(b_2, 1)$, where $b_1 = 1/2$ or $b_2 = 1/2$, on each of which the values of $\mathcal{P}_m^{u_m+1}(g_m^\star)$ are contained in an subinterval of $(0, \infty)$ of length $32\mu/C$. Here $C$ is the normalization constant from equation (4.4). From $f_{c,m} < 3/2$ and $\mu < 1/4$ we get $g_m(y) < 2$ and $\mathcal{P}_m^{u_m}(g_m^\star)(x) \leq 2/(c_0 c_{u_m}) \leq 32$, since we deduced earlier from equation (4.18) that $c_0 c_{u_m} \geq 1/16$. For $m$ large enough, say $m \geq M_2 > M_1$, we will have $\ell(\varepsilon/m) + \ell(\delta/m) < 1/64$, where $\ell$ is defined as in equation (4.2). Considering the integral of $\mathcal{P}_m^{u_m}(g_m^\star)$ over the subset of [0,1] mapped outside the unit interval by $F_m$, we obtain a lower bound of $1 - 32/64 = 1/2$ on $C$. So the values of $\mathcal{P}_m^{u_m+1}(g_m^\star)$ on each of $(0, b_1)$, $(b_1, b_2)$, $(b_2, 1)$ are contained in intervals of length $64\mu$. Using (4.4), calculations show that we can find a bound, independent of choice of $g_m$ with $\|g_m - f_{c,m}\|_\infty < 1/4$, on the range of values of $\mathcal{P}_m^{u_m+1}(g_m^\star)$ over all of $(0, 1/2)$ and $(1/2, 1)$ respectively. Call this bound $S$. We now may choose a piecewise constant map $k_m \in K_S$, as defined in Definition 4.8, so that

$$\left\|\mathcal{P}_m^{u_m}(g_m^\star) - k\right\|_\infty < 32\mu.$$

Recalling that $f_{c,m} > 1/2$, we get $g_m > 1/4$, and equation (4.17) gives us lower bound of $1/4$ on $\mathcal{P}_m^{u_m}(g_m^\star)$ and by equation (4.4) a lower bound on $\mathcal{P}_m^{u_m+1}(g_m^\star)$, which could be taken for example as $1/20$. We then apply Corollary 4.10 with $f = \mathcal{P}_m^{u_m+1}(g_m^\star)$, $g = k_m$, $s = 1/20$ and to find $L$ such that for $n \geq 1$ we have

$$\left\|\mathcal{P}_m^{u_m+n}(g_m^\star) - \mathcal{P}_m^n(k_m)\right\| < 32\mu L, \tag{4.19}$$

regardless of choice of $g_m$. By lemma 4.11 there exists $N$ such that for all $m \geq M_2$ and for $n \geq N$ we have $\left\|\mathcal{P}_m^n(k_m) - f_{c,m}\right\|_\infty < \mu$. Take $B = 32L + 1$ and $M = M_2$ to obtain equation (4.15). ∎

**Lemma 4.15.** *Let $A_m$ denote the subset of [0,1] mapped outside of [0,1] by $F_m$. Take $0 < \mu < 1/4$ and let $g_m$ be a density with $\|g_m - f_{c,m}\|_\infty < \mu$. Let $V_m$ be distributed according to that density and $g^\star$ denote the density corresponding to the distribution of $F_r(V_m; \varepsilon/m, \delta/m)$, conditional on $V_m \in A_m$. Then there exists $B > 0$, $M \in \mathbb{N}$ and $N \in \mathbb{N}$ such that for all $m \geq M$ and $n \geq N + \lceil \log(m/\varepsilon)/\log(4 + \varepsilon/m)\rceil$ and for any sequence of $g_m$ satisfying above properties we have*

$$\left\|\mathcal{P}_m^n(g_m^\star) - f_{c,m}\right\|_\infty < \mu B. \tag{4.20}$$

*Proof.* We partition on the events $V_m \in (0, 1/4)$, $V_m \in (1/4, 1/2)$, $V_m \in (1/2, 3/4)$ and $V_m \in (3/4, 1)$, then apply the same arguments as for $V_m \in (0, 1/4)$ in lemma 4.14. ∎

**Lemma 4.16.** *Let $\delta$, $\varepsilon > 0$. Let $0 < 1/2 < d$. For $m \in \mathbb{N}$ let $Y'_m(t)$ and $T'_{m,j}$, $j = 1, 2, \ldots$, be as described in definition 4.12. Then for any $k \geq 1$ and $0 < t_1 < \cdots < t_k$ we have*

$$\mathbb{P}\left(T'_{m,k+1} \leq \frac{\lfloor mt_k\rfloor}{m} + \tau \mid T'_{m,k} = \frac{\lfloor mt_k\rfloor}{m}, \ldots, T'_{m,1} = \frac{\lfloor mt_1\rfloor}{m}\right) \to 1 - \exp(-\gamma\tau)$$

*as $m \to \infty$, where $\gamma$ is given in equation (4.3).*

*Proof.* We first describe how the density corresponding to $F_r^{\lfloor mt\rfloor}(X'_m; \varepsilon/m, \delta/m)$ develops, conditional on $T'_{m,k} = \lfloor mt_k\rfloor/m, \ldots, T'_{m,1} = \lfloor mt_1\rfloor/m$. Let $\mu > 0$. Using equation (4.14), there exist $N_1 \in \mathbb{N}$ and $M_1 \in \mathbb{N}$ so that $n \geq N_1$ and $m \geq M_1$ implies $\left\|\mathcal{P}_m^n(f'_{i,m}) - f_{c,m}\right\|_\infty \leq \mu B(d)$. For large enough $m$, we will have $\lfloor mt_1\rfloor > N_1$. Choosing $\mu$ small enough, we will have $\mu B(d) < 1/4$ and so can apply lemma 4.15 with $g_m = \mathcal{P}_m^{\lfloor mt_1\rfloor - 1}(f'_{i,m})$. Write $g_m^{(1)} = g_m^\star$ for the density after the jump at time $\lfloor mt_1\rfloor/m$. There exists $N \in \mathbb{N}$ and $B > 0$ such that for large enough $m$ and $n \geq N + \lceil \log(m/\varepsilon)/\log(4 + \varepsilon/m)\rceil = N + S(m)$

$$\left\|\mathcal{P}_m^n(g_m^{(1)}) - f_{c,m}\right\|_\infty < \mu B(d)B.$$

Between time $\lfloor t_j m \rfloor / m$ and $\lfloor t_{j+1} m \rfloor / m$, the density of $F_r^{\lfloor mt \rfloor}(X'_m; \varepsilon/m, \delta/m)$ develops as given by applying operator $\mathcal{P}_m$. Provided that $\mu < B^{-k}/4$ and $m$ is large enough so that $N + S(m) < \lfloor mt_{j+1} \rfloor - \lfloor mt_j \rfloor$ for $j = 1$, $2, \ldots, k$, we can iteratively apply lemma 4.15 with $g_m = \mathcal{P}_m^{\lfloor mt_{j+1} \rfloor - \lfloor mt_j \rfloor - 1}(g_m^{(j)})$, where $g_m^{(j)}$ is the density after the $j$-th jump has occurred, at time $\lfloor mt_j \rfloor / m$. Then for large enough $m$ and $n \geq N + S(m)$ we in particular find

$$||\mathcal{P}_m^n(g_m^{(k)}) - f_{c,m}||_\infty \leq \mu B(d) B^k. \tag{4.21}$$

But $\mathcal{P}_m^n(g_m^{(k)})$ describes the densities of $F_r^{\lfloor mt \rfloor}(X'_m; \varepsilon/m, \delta/m)$ conditional on $T'_{m,k} = \lfloor mt_k \rfloor / m, \ldots,$ $T'_{m,1} = \lfloor mt_1 \rfloor / m$ and no further jump occurring. Choose $\tau > 0$. Then

$$\mathbb{P}\left(T_m^k > t_k + \tau \,\middle|\, T_m^k = \frac{\lfloor mt_k \rfloor}{m}, \ldots, T_m^1 = \frac{\lfloor mt_1 \rfloor}{m}\right) = \prod_{n=1}^{\lfloor m\tau \rfloor} \int_{[0,1] \setminus A_m} \mathcal{P}_m^{n-1}(g_m^{(k)})(x) \, dx,$$

where $A_m$ is defined as in lemma 4.15. Since equation (4.21) holds, we can use the same arguments as in the proof of lemma 4.13 to find lower and upper bounds on

$$\prod_{n=S(m)+1}^{\lfloor m\tau \rfloor} \int_{[0,1] \setminus A_m} \mathcal{P}_m^{n-1}(g_m^{(k)})(x) \, dx$$

converging to $\exp[-\gamma(1 + \mu B(d)B)\tau]$ and $\exp[-\gamma(1 - \mu B(d)B)\tau]$, respectively, as $m \to \infty$. Using equation (4.19) from lemma 4.14, there are $l_m \in K_S$ and $L > 0$ such that

$$||\mathcal{P}_m^{u_m+n}(g_m^{(k)}) - \mathcal{P}_m^n(l_m)||_\infty < 32 \mu B(d) B^k L,$$

for $n \geq 1$, implying that $\mathcal{P}_m^n(g_m^{(k)})$ stays close to a piecewise constant function in $K_S$ as soon as jumps are possible. This tells us that an upper bound $b$ on the densities $\mathcal{P}_m^n(g_m^{(k)})$ can be found, valid for all $n \geq u_m$ and all $m$ large enough. But then

$$\left(1 - b\left(\ell\left(\frac{\varepsilon}{m}\right) + \ell(\delta/m)\right)\right)^{S(m)} \leq \prod_{n=1}^{S(m)} \int_{[0,1] \setminus A_m} \mathcal{P}_m^{n-1}(g_m^{(k)})(x) \, dx \leq 1.$$

The expression on the left converges to 1 as $m \to \infty$ since $S(m)$ grows like log. But combining this with our earlier bounds with limits $\exp[-\gamma(1 \pm \mu B(d)B)\tau]$ and letting $\mu \to 0$, we find that

$$\mathbb{P}\left(T'_{m,k+1} \leq \frac{\lfloor mt_k \rfloor}{m} + \tau \,\middle|\, T'_{m,k} = \frac{\lfloor mt_k \rfloor}{m}, \ldots, T'_{m,1} = \frac{\lfloor mt_1 \rfloor}{m}\right) \to 1 - \exp(-\gamma\tau) \quad \text{as } m \to \infty. \quad \blacksquare$$

Theorem 4.4 is now simply a corollary of lemma 4.16. Let $Y_m(t)$ and $Y'_m(t)$ be as defined in theorem 4.4 and definition 4.12 respectively, recalling that $Y'_m$ depends on a choice of $0 < d < 1/2$. Write $E$ and $E'$ for events $(T_{m,k} = \lfloor mt_k \rfloor / m, \ldots, T_{m,1} = \lfloor mt_1 \rfloor / m)$ and $(T'_{m,k} = \lfloor mt_k \rfloor / m, \ldots, T'_{m,1} = \lfloor mt_1 \rfloor / m)$, respectively. By definition of initial distributions of $Y_m$ and $Y'_m$, $X_m$ and $X'_m$, with underlying densities $f_{i,m}$ and $f'_{i,m}$, we have that

$$\mathbb{P}\left(T_{m,k+1} \leq \frac{\lfloor mt_k \rfloor}{m} + \tau \,\middle|\, E, X_m \in [d, 1-d]\right) = \mathbb{P}\left(T'_{m,k+1} \leq \frac{\lfloor mt_k \rfloor}{m} + \tau \,\middle|\, E', X'_m \in [d, 1-d]\right).$$

We have already noted earlier, in the proof of lemma 4.13, that $\sup_m ||f_{i,m}||_\infty < \infty$ and $\sup_m ||f'_{i,m}||_\infty < \infty$. Conditioning on the events $X_m, X'_m \in [d, 1-d]$ and $X_m, X'_m \notin [d, 1-d]$, we get

$$\left|\mathbb{P}\left(T_{m,k+1} \leq \frac{\lfloor mt_k \rfloor}{m} + \tau \,\middle|\, E\right) - \mathbb{P}\left(T'_{m,k+1} \leq \frac{\lfloor mt_k \rfloor}{m} + \tau \,\middle|\, E'\right)\right|$$

$$\leq \mathbb{P}(X_m \notin [d, 1-d]) + \mathbb{P}(X'_m \notin [d, 1-d]) \leq 2d\left(\sup_m ||f_{i,m}||_\infty + \sup_m ||f'_{i,m}||_\infty\right).$$

Since $0 < d < 1/2$ was arbitrary, we apply lemma 4.16 and let $d \to 0$ to conclude the proof. $\square$

# 5. Discussion

We have seen in §3 that the behaviour of the trajectories of a shift-periodic map $F$ with integer spikes (definition 3.2) can be described in terms of discrete-time random walks for suitable initial distributions, and we observed a variety of interesting stochastic processes in scaling limits. Omitting the integer spike condition, the class of maps considered appears to be too large to allow for non-trivial statements about trajectories which apply to all shift-periodic maps. Some of the potential

issues (such as complicated expressions for invariant densities) can be seen in the proofs of §4. Nevertheless, this section demonstrates that interesting stochastic behaviour can occur for general shift-periodic maps. The proof strategies of §4 can also be extended to other parameter-dependent families of shift-periodic maps with small holes. For instance, if $F(x; \varepsilon/m, \delta/m)$ is replaced in theorem 4.4 by a sequence of shift-periodic maps $F_m : \mathbb{R} \to \mathbb{R}^\infty$ such that the conditionally invariant density of $F_m$ on interval [0,1] converges uniformly to 1 as $m \to \infty$, and they satisfy $\lambda\{x \in [0,1] : F_m(x) \notin [0,1]\} \to 0$ and $m\lambda\{x \in [0,1] : F_m(x) \notin [0,1]\} \to \gamma$ as $m \to \infty$, we again obtain behaviour like that of a continuous-time random walk in a limit, with waiting times distributed according to an exponential distribution with mean $1/\gamma$.

Data accessibility. This article has no additional data.

Competing interests. We declare we have no competing interests.

Authors' contributions. Both authors equally contributed to the study and to the draft of the manuscript. Both authors gave final approval for publication.

Funding. This work was supported by the Royal Society.

Acknowledgements. R.E. thank the Royal Society for a University Research Fellowship.

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
