## [Reviewer comments · Royal Society Open Science]

Review History

RSOS-191423.R0 (Original submission)

Review form: Reviewer 1

Is the manuscript scientifically sound in its present form?

Yes

Are the interpretations and conclusions justified by the results?

Yes

Is the language acceptable?

Yes

Do you have any ethical concerns with this paper?

No

Have you any concerns about statistical analyses in this paper?

No

Recommendation?

Accept as is

Comments to the Author(s)

The authors have further expanded the revised version, and added some specific physics literature to the list of references, since one of the referees was insisting on this. Overall I think the paper is nice and interesting, and the theorems proved are useful to know. I recommend acceptance.

Decision letter (RSOS-191423.R0)

28-Oct-2019

Dear Dr Erban:

It is a pleasure to accept your manuscript entitled "Limiting stochastic processes of shift-periodic dynamical systems" in its current form for publication in Royal Society Open Science. The comments of the reviewer(s) who reviewed your manuscript are included at the foot of this letter.

Kind regards,
Anita Kristiansen
Editorial Coordinator
Royal Society Open Science
openscience@royalsociety.org

on behalf of Dr Jose Carrillo (Associate Editor) and Professor Mark Chaplain (Subject Editor).

Reviewer(s)' Comments to Author:

Reviewer: 1

Comments to the Author(s)

The authors have further expanded the revised version, and added some specific physics literature to the list of references, since one of the referees was insisting on this. Overall I think the paper is nice and interesting, and the theorems proved are useful to know. I recommend acceptance.
